# CONSISTENCY MODELS AS A RICH AND EFFICIENT POLICY CLASS FOR REINFORCEMENT LEARNING

**Zihan Ding, Chi Jin**
Department of Electrical and Computer Engineering
Princeton University
`{zihand, chij}@princeton.edu`

## ABSTRACT

Score-based generative models like the diffusion model have been testified to be effective in modeling multi-modal data from image generation to reinforcement learning (RL). However, the inference process of diffusion model can be slow, which hinders its usage in RL with iterative sampling. We propose to apply the consistency model as an efficient yet expressive policy representation, namely *consistency policy*, with an actor-critic style algorithm for three typical RL settings: offline, offline-to-online and online. For offline RL, we demonstrate the expressiveness of generative models as policies from multi-modal data. For offline-to-online RL, the consistency policy is shown to be more computational efficient than diffusion policy, with a comparable performance. For online RL, the consistency policy demonstrates significant speedup and even higher average performances than the diffusion policy.

## 1 INTRODUCTION

Parameterized policy representation is an important component for policy-based deep reinforcement learning (DRL) (Sutton & Barto, 2018; Arulkumaran et al., 2017; Dong et al., 2020). Prior works have developed a variety of policy parameterization methods. For discrete action space, existing policy parameterization includes Softmax action preferences (Sutton & Barto, 2018), Gumbel-Softmax for categorical distributions (Jang et al., 2016), decision trees (Frosst & Hinton, 2017; Ding et al., 2020), etc. For continuous action space, the most typical choice is unimodal Gaussian distribution. However, in practice the demonstration dataset often encompasses samples from a mixture of behavior policies. To capture the multi-modality in data distribution, Gaussian mixture model (GMM) (Jacobs et al., 1991; Ren et al., 2021), variational auto-encoders (VAE) (Kingma & Welling, 2013; Kumar et al., 2019), denoising diffusion probabilistic model (DDPM) (Ho et al., 2020; Song et al., 2020; Wang et al., 2022; Chi et al., 2023; Hansen-Estruch et al., 2023; Venkatraman et al., 2023) are adopted as policy representation.

The desiderata for policy representation in DRL includes: 1. The strong expressiveness of the function class is found to be critical for modeling multi-modal data distribution in offline RL (Wang et al., 2022) or imitation learning (IL) (Chi et al., 2023); 2. Differentiability of the model is usually required for ease of optimization with stochastic gradient descent; 3. Computational and time efficiency for sampling can be essential for RL agents learning from interactions with environments. Previous works with action diffusion models (*i.e.*, diffusion policy) testify the expressiveness of diffusion models for multi-modal action distributions (Wang et al., 2022; Chi et al., 2023; Hansen-Estruch et al., 2023; Janner et al., 2022; Ajay et al., 2022). Although GMM and VAE also capture multi-modality, diffusion models with large sampling steps are found to be more expressive for IL and offline RL scenarios (Wang et al., 2022; Chi et al., 2023). However, it is known that the diffusion model with progressive denoising over a large number of steps can lead to slow sampling speed. The action inference can be a critical bottleneck for online RL heavily depends on sampling from environments. A direct usage of *diffusion policies for online settings with policy gradient for optimization requires backpropagating through the diffusion networks for the number of sampling steps*, which is not scalable for its *large time consumption and memory occupancy*. Consistency models (Song et al., 2023) based on probability flow ordinary differential equation (ODE) is proposed as a rescue with comparable performances as diffusion models but much less computational time, which allows few-step generation process thus significantly reduce the time consumption at inference stage.

This paper takes the first step adapting the consistency model–an expressive yet efficient generative model–as policy representation for DRL. The consistency policy is embedded in both behavioral

cloning (BC) method and an actor-critic (AC) algorithm, namely Consistency-BC and Consistency-AC. Experimental evaluation includes three typical RL settings: offline, offline-to-online and online. Policies with two generative models–diffusion model and consistency model–are thoroughly compared in all three settings on D4RL dataset (Fu et al., 2020). For offline RL, we propose a new loss scaling for stabilizing the training process of consistency policy with policy regularization, and demonstrate the expressiveness of two generative policy models. This is illustrated by showing BC with an expressive model like diffusion or consistency provides fairly good policies outperforming some previous offline RL methods. The performances are further improved by leveraging the actor-critic style algorithm with necessary policy regularization to avoid generating out-of-distribution actions. The fast sampling process of the consistency policy not only helps to reduce the training time, *e.g.*, by $43\%$ for offline BC, but more importantly, improves the time efficiency for online interaction in the environments by accelerating action inference. For offline-to-online setting with initialized models trained on offline dataset and online setting with learning from scratch, the consistency policy shows comparable or even higher performances than the diffusion policy in some tasks, using significantly shorter wall-clock time for training and inference. The source code is available[1].

## 2   RELATED WORKS

**Offline and Offline-to-Online RL.**   The *offline* RL is the problem of policy optimization with a fixed dataset. It is well known for suffering from the value overestimation problem for out-of-distribution states and actions from the dataset. Existing methods for solving this issue fall into categories of (1) explicitly constraining the learning policy with offline data using batch constraining, behavior cloning (BC) or divergence constraints (*e.g.*, Kullback-Leibler, maximum mean discrepancy), including algorithms Batch-Constrained deep Q-learning (BCQ) (Fujimoto et al., 2019), TD3+BC (Fujimoto & Gu, 2021), Onestep RL (Brandfonbrener et al., 2021), Advantage Weighted Actor-Critic (AWAC) (Nair et al., 2020), Bootstrapping Error Accumulation Reduction (BEAR) (Kumar et al., 2019), BRAC (Wu et al., 2019), Diffusion Q-learning (Diffusion QL) (Wang et al., 2022), Extreme Q-learning ($\mathcal{X}$-QL) (Garg et al., 2023) and Actor-Restricted Q-learning (ARQ) (Goo & Niekum, 2022), or (2) implicit regularization with pessimistic value estimation, like Conservative Q-learning (CQL) (Kumar et al., 2020), Random Ensemble Mixture (REM) (Agarwal et al., 2020), Implicit Q-learning (IQL) (Kostrikov et al., 2021), Implicit Diffusion Q-learning (IDQL) Hansen-Estruch et al. (2023), Model-based Offline Policy Optimization (MOPO) (Yu et al., 2020), etc. MoRel (Kidambi et al., 2020) is a model-based offline RL algorithm constructing pessimistic MDP for learning conservative policies, which does not clearly fall into above two categories. *Offline-to-online* RL usually suffers from a catastrophic degraded performance at initial online training stage, due to the distribution shift of training samples. Previous research has studied online fine-tuning with offline data or pre-trained policies, including Hybrid Q learning (Song et al., 2022), RLPD (Ball et al., 2023), Cal-QL (Nakamoto et al., 2023), Action-free Guide (Zhu et al., 2023), Actor-Critic Alignment (ACA) (Yu & Zhang, 2023) and Lee et al. (2022).

**Score-based Generative Model for RL.**   For policy representation in RL, recent work also uses Denoising Diffusion Probabilistic Models (DDPM) (Ho et al., 2020; Song et al., 2020), which we loosely refer to as the diffusion model (original diffusion model traces back to Sohl-Dickstein et al. (2015)) in this paper, to capture the multi-modal distributions in offline dataset. Diffusion QL (Wang et al., 2022) uses diffusion model for policy representation in the Q-learning+BC approach. Implicit Diffusion Q-learning (IDQL) (Hansen-Estruch et al., 2023) is a variant of IQL using diffusion policy. Diffusion policies (Chi et al., 2023) applies diffusion models for policy representation under imitation learning settings in robotics domain. Diffuser (Janner et al., 2022) and Decision Diffuser (Ajay et al., 2022) combines decision transformer architecture with diffusion models for model-based reinforcement learning from offline dataset. Diffusion policies are also used for goal-conditioned imitation learning (Reuss et al., 2023) and human behavior imitation (Pearce et al., 2023). Q-guided Policy Optimization (QGPO) (Lu et al., 2023) proposes a new formulation for intermediate guidance in diffusion sampling process. Latent Diffusion-Constrained Q-Learning (LDCQ) (Venkatraman et al., 2023) proposes to apply latent diffusion model with a batch-constrained Q value to handle the stitching issue and the extrapolation errors for offline dataset.

---

[1] https://github.com/quantumiracle/Consistency_Model_For_Reinforcement_Learning

## 3 PRELIMINARIES

### 3.1 OFFLINE AND ONLINE RL

For RL, we define a Markov decision process $(\mathcal{S},\mathcal{A},R,\mathcal{T},\rho_0,\gamma)$, where $\mathcal{S}$ is the state space, $\mathcal{A}$ is the action space, $R(s,a):\mathcal{S}\times\mathcal{A}\to\mathbb{R}$ is the reward function, $\mathcal{T}(s'|s,a):\mathcal{S}\times\mathcal{A}\to\mathrm{Pr}(\mathcal{S})$ is the stochastic transition function, $\rho_0(s_0):\mathcal{S}\to\mathrm{Pr}(\mathcal{S})$ is the initial state distribution, and $\gamma\in[0,1]$ is the discount factor for value estimation. A stochastic policy $\pi(a|s):\mathcal{S}\to\mathrm{Pr}(\mathcal{A})$ determines the action $a\in\mathcal{A}$ for the agent to take given its current state $s\in\mathcal{S}$, and the optimization objective for the policy is its discounted cumulative reward: $\mathbb{E}_\pi[\sum_{t=0}^\infty\gamma^t r(s_t,a_t)]$. For offline RL, there exist a dataset $\mathcal{D}=\{(s,a,r,s',\text{done})\}$ collected with some behavior policies $\pi^b$, and the current policy $\pi$ is set to be optimized with $\mathcal{D}$. For online RL, the agent is allowed collect samples through interacting with the environment to compose an online training dataset $\tilde{\mathcal{D}}$ for optimizing its policy. We consider parameterized policy representation as $\pi_\theta$.

### 3.2 CONSISTENCY MODEL

The diffusion model (Ho et al., 2020; Song et al., 2020) solves the multi-modal distribution matching problem with a stochastic differential equation (SDE), while the consistency model (Song et al., 2023) solves an equivalent probability flow ordinary differential equation (ODE): $\frac{d\mathbf{x}_\tau}{d\tau}=-\tau\nabla\log p_\tau(\mathbf{x})$ with $p_\tau(\mathbf{x})=p_{\text{data}}(\mathbf{x})\otimes\mathcal{N}(\mathbf{0},\tau^2\mathbf{I})$ for time period $\tau\in[0,T]$, where $p_{\text{data}}(\mathbf{x})$ is the data distribution. The reverse process along the solution trajectory $\{\hat{\mathbf{x}}_\tau\}_{\tau\in[\epsilon,T]}$ of this ODE is the data generation process from initial random samples $\hat{\mathbf{x}}_T\sim\mathcal{N}(\mathbf{0},T^2\mathbf{I})$, with $\epsilon$ as a small constant close to $0$ for handling numerical problem at the boundary. For speeding up the sampling process from a diffusion model, consistency model shrinks the required number of sampling steps to a much smaller value than the diffusion model, without hurting the model generation performance much. Specifically, it approximates a parameterized consistency function $f_\theta:(\mathbf{x}_\tau,\tau)\to\mathbf{x}_\epsilon$, which is defined as a map from the noisy sample $\mathbf{x}_\tau$ at step $\tau$ back to the original sample $\mathbf{x}_\epsilon$, instead of applying a step-by-step denoising function $p_\theta(\mathbf{x}_{\tau-1}|\mathbf{x}_\tau)$ as the reverse diffusion process in diffusion model. The training and inference details of consistency model refer to Appendix B. For modeling the conditional distribution with condition variable $c$, the consistency function is changed to be $f_\theta(c,\mathbf{x}_\tau,\tau)$, which is sightly different from original consistency model.

## 4 CONSISTENCY MODEL AS RL POLICY

The consistency model as policy representation in RL can be formulated in the following way. To map the consistency model to a policy in MDP, we set:

$$c\triangleq s,\quad \mathbf{x}\triangleq a,\quad p_{\text{data}}(\mathbf{x})\triangleq p_\mathcal{D}(a|s),\quad \pi_\theta(s)\triangleq\texttt{Consistency\_Inference}(s;f_\theta) \tag{1}$$

where $p_\mathcal{D}(a|s)$ is the action-state conditional distribution from offline dataset $\mathcal{D}$.

**Consistency Action Inference.** By setting the condition variable $c$ as state $s$ and generated variable $\mathbf{x}$ as action $a$, the consistency function $f_\theta$ can be used for generating actions from states following the conditional distribution of the dataset, *i.e.*, a behavior RL policy. The parameterized policy $\pi_\theta$ is defined implicitly in terms of $f_\theta$, with which an action $a$ conditioned on state $s$ can be generated following the $\texttt{Consistency\_Inference}$ as Alg. 1 with predetermined $\{\tau_n|n\in[N]\}$ sequence. During the inference process, a trained consistency model $f_\theta(s,\hat{a}_{\tau_n},\tau_n)$ iteratively predicts denoised samples from the noisy inputs $\hat{a}_{\tau_n}=a+\sqrt{\tau_n^2-\epsilon^2}z$ along the probability flow ODE trajectory at step $n\in[N]$, with Gaussian noise $z\sim\mathcal{N}(\mathbf{0},\mathbf{I})$. $\{\tau_n|n\in[N]\}$ is a sub-sequence of time points on a certain time period $[\epsilon,T]$ with $\tau_1=\epsilon,\tau_N=T$. For inference, the sub-sequence is a linspace of $[\epsilon,T]$ with $(N-1)$ sub-intervals. A single-step version of $\texttt{Consistency\_Inference}$ can be achieved by just set $\{\tau_n|n=0,1\}=\{\epsilon,T\}$. Notice that $T$ here is the time horizon for denoising process in the consistency model instead of the episode length of the sample trajectory.

**Consistency Behavior Cloning.** With the offline dataset $\mathcal{D}$, the conditional consistency model as policy can be trained with loss by adapting the original (Song et al., 2023):

$$\mathcal{L}_c(\theta)=\mathbb{E}_{n\sim\mathcal{U}(1,N-1),(s,a)\sim\mathcal{D},z\sim\mathcal{N}(\mathbf{0},\mathbf{I})}\Big[\lambda(\tau_n)d\big(f_\theta(s,a_{\tau_{n+1}},\tau_{n+1}),f_{\theta^\top}(s,a_{\tau_n},\tau_n)\big)\Big] \tag{2}$$

where $\lambda(\cdot)$ is a step-dependent weight function, $a_{\tau_n}=a+\tau_n z$ and $d(\cdot,\cdot)$ is the distance metric. $f_{\theta^\top}$ is exponential moving average of $f_\theta$ for stabilizing the target estimation in training. In classical actor-critic algorithm, there exists the same delayed update of the policy network $\pi_{\theta^\top}$ (*i.e.*, $f_{\theta^\top}$) for estimating target $Q$-values, which is set to coincide with the target in estimating the consistency loss. The setting for $\tau_n$ is detailed in Appendix B. Pseudo-code of Consistency BC refers to Alg. 2.

---

**Algorithm 1** Consistency Action Inference

---

**Input** $s, f_\theta, N, \{\tau_n\}_{n \in [N]}$
Initial $a \leftarrow f_\theta(s, \hat{a}_T, T), \hat{a}_T \sim \mathcal{N}(\mathbf{0}, T^2\mathbf{I})$
**for** $n = N-1$ to $2$ **do**
    $\hat{a}_{\tau_n} \leftarrow a + \sqrt{\tau_n^2 - \epsilon^2} z, z \sim \mathcal{N}(\mathbf{0}, \mathbf{I})$
    $a \leftarrow f_\theta(s, \hat{a}_{\tau_n}, \tau_n)$
**end for**
**return** $a$

---

**Algorithm 2** Consistency Behavior Cloning

---

**Input** offline dataset $\mathcal{D}$
Initialize consistency policy $\pi_\theta$, target $\theta^\mathsf{T} \leftarrow \theta$
**for** iterations $k = 1, ..., K$ **do**
    Update policy $\pi_\theta$ (with model $f_\theta$) using loss $\mathcal{L}_c(\theta)$ as Eq. 2;
    Update target: $\theta^\mathsf{T} \leftarrow \alpha\theta^\mathsf{T} + (1-\alpha)\theta$
**end for**

---

**Algorithm 3** Offline Consistency Actor-Critic

---

**Input** offline dataset $\mathcal{D}$
Initialize consistency policy network $\pi_\theta$, critic networks $Q_{\phi_1}, Q_{\phi_2}$
Initialize target network parameters: $\theta^\mathsf{T} \leftarrow \theta$, $\phi_1^\mathsf{T} \leftarrow \phi_1, \phi_2^\mathsf{T} \leftarrow \phi_2$
**for** policy training iterations $k = 1, ..., K$ **do**
    Sample minibatch $\mathcal{B} = \{(s, a, r, s')\} \subseteq \mathcal{D}$;
    % Q-value Update
    Update $Q_{\phi_1}, Q_{\phi_2}$ with Eq. 3;
    % Policy Update
    Update policy $\pi_\theta$ (with model $f_\theta$) via Eq. 4;
    % Target Update
    Update target: $\theta^\mathsf{T} \leftarrow \alpha\theta^\mathsf{T} + (1-\alpha)\theta, \phi_i^\mathsf{T} \leftarrow \alpha\phi_i^\mathsf{T} + (1-\alpha)\phi_i, i \in \{1, 2\}$;
**end for**
**return** $\pi_\theta, Q_{\phi_1}, Q_{\phi_2}$

---

**Consistency Actor-Critic.** As as estimation of the state-action value of current policy, the parameterized $Q_\phi(s, a)$ function can be learned with the double Q-learning loss (Fujimoto et al., 2018) with batched data $\mathcal{B} \subseteq \mathcal{D}$:

$$\mathcal{L}(\phi) = \mathbb{E}_{(s,a,s') \sim \mathcal{B}, a' \sim \pi_{\theta^\mathsf{T}}(\cdot | s')} \left[ \left( \left( r(s,a) + \gamma \min_{i \in \{1,2\}} Q_{\phi_i^\mathsf{T}}(s', a') \right) - Q_{\phi_i}(s,a) \right)^2 \right] \tag{3}$$

with $Q_{\phi_i^\mathsf{T}}$ as a delayed update of $Q_{\phi_i}, i \in \{1, 2\}$ for stabilizing training.

The regularized policy $\pi_\theta$ on offline dataset is learned with a combination of policy gradient through maximizing the expected $Q_\phi(s, a)$ function and a behavior cloning regularization with consistency loss $\mathcal{L}_c(\theta)$:

$$\mathcal{L}(\theta) = \mathcal{L}_c(\theta) + \eta \mathcal{L}_q(\theta) \tag{4}$$

$$\text{where } \mathcal{L}_q(\theta) = -\mathbb{E}_{s \sim \mathcal{B}, a \sim \pi_\theta(s)} \left[ Q_\phi(s, a) \right] \tag{5}$$

where $a \sim \pi_\theta(s)$ is action inference from the consistency policy as Alg.1. It can be noticed that the actions generated with $N$ denoising steps will produce the policy gradients through the $Q_\phi(s, a)$ in above equation, thus it also backpropagates through $f_\theta$ for $N$ times in the gradient descent procedure, which can lead to additional time consumption apart from the multi-step model inference. Therefore, reducing the denoising steps $N$ can be critical for the speed of this type of models as RL policies. The consistency actor-critic (Consistency-AC) algorithm is provided in pseudo-code Alg. 3.

**Loss Scaling.** The consistency loss as Eq. 2 matches the denoised predictions from two consecutive timesteps $\tau_n$ and $\tau_{n+1}$. Due to the usage of $N(k)$ schedule (detailed in Appendix B), their difference $|\tau_{n+1} - \tau_n|$ decreases as the training iteration $k$ increases (thus $N(k)$ also increases), which allows the consistency model to have a coarse-to-fine matching process across different time scales. This also leads to a decreasing loss value $\mathcal{L}(\theta; k)$ as $k$ increases from 1 to $K$ since the predictions from smaller time intervals are easier to match. Actually, the loss $\mathcal{L}_c(\theta)$ changes drastically across several magnitudes within an epoch, which leads to severe imbalance with the second loss term $\mathcal{L}_q(\theta)$ in Eq. 4. The coefficient $\eta$ is a constant hyperparameter independent of $k$, so it cannot help to alleviate this issue. Although original consistency model applies $\lambda(\tau_n) \equiv 1$ for image generation, we empirically find that in offline RL this imbalance of two loss terms can hurt the effect of policy regularization in some tasks, as evidenced by ablation studies in Sec. 5.2. To solve this issue, we propose a $k$-dependent weighting mechanism to balance the values of two loss terms. This is found to improve the performances of this policy regularization method with consistency model on offline RL. Specifically, $\lambda(\cdot)$ in Eq. 2 is chosen to be: $\lambda(\tau_n, \tau_{n+1}; k) = \frac{\xi}{|\tau_{n+1}(k) - \tau_n(k)|}$ where $\xi$ is set according to tasks (or absorbed in $\eta$). The denominator captures the loss scale at iteration $k$ conveniently.

## 5 EXPERIMENTAL EVALUATION

To evaluate the expressiveness and computational efficiency of the proposed consistency policy and corresponding algorithms, we conduct experiments on four task suites (Gym, AntMaze, Adroit, Kitchen) in

D4RL benchmarks under three canonical RL settings: offline (Sec. 5.1, Sec. 5.2), offline-to-online and online (Sec. 5.3). It is known that the D4RL offline dataset can exhibit multi-modality since the samples may be collected with a mixture of polices or along various sub-optimal trajectories, which makes the expressiveness of policy representation critical (Fu et al., 2020; Wang et al., 2022). For offline RL, the generative models as policies are evaluated with both behavior cloning (Consistency-BC, Diffusion-BC) and actor-critic type (Consistency-AC, Diffusion-QL) algorithms, in terms of both performances and computational time. The Diffusion-QL is also an actor-critic algorithm though with name QL. Variants of Consistency-AC are compared as ablation studies and the best performances are reported. For offline-to-online and online RL settings, the learning curves and final results are compared for different methods. For evaluation, each model is evaluated over 10 episodes for Gym tasks and 100 episodes for other tasks, following the settings in previous work (Wang et al., 2022). By default, the consistency policy applies the number of denoising steps $N = 2$ with a saturated performances on most of D4RL tasks, while diffusion policy uses $N = 5$ (Wang et al., 2022). Effects of different choices of $N$ are discussed in Sec. 5.2.

## 5.1 OFFLINE RL: BEHAVIOR CLONING WITH EXPRESSIVE POLICY REPRESENTATION

**Empirical finding 1:** *By behavior cloning alone (without any RL component), using an expressive policy representation with multi-modality like the consistency or diffusion model achieves performances comparable to many existing popular offline RL methods. Learning consistency policy requires much less computation than learning diffusion policy.*

The proposed method **Consistency-BC** follows Alg. 2 with consistency policy for behavior cloning, and **Diffusion-BC** is by replacing the policy representation with a diffusion model and replace the policy loss with the diffusion model training loss $\mathcal{L}_d(\theta)$ as specified in paper (Wang et al., 2022). Results for classic BC with Gaussian policies and previous offline RL baselines, including AWAC (Nair et al., 2020), Diffuser (Janner et al., 2022), MoRel (Kidambi et al., 2020), Onestep RL (Brandfonbrener et al., 2021), TD3+BC (Fujimoto & Gu, 2021), Decision Transformer (DT) (Chen et al., 2021), BCQ (Fujimoto et al., 2019), BEAR (Kumar et al., 2019), BRAC (Wu et al., 2019) and REM (Agarwal et al., 2020), are adopted from previous paper (Wang et al., 2022). SAC (Haarnoja et al., 2018) is the

Table 1: The average scores of vanilla BC (with Gaussian), Consistency-BC, Diffusion-BC and several offline RL baselines on D4RL Gym, AntMaze, Adroit, and Kitchen tasks are shown. For Consistency-BC and Diffusion-BC, each cell has two values: one for offline model selection and another (in brackets) for online model selection. Each result is averaged over five random seeds with standard deviations reported. The bold values are the highest among each row.

| Gym Tasks | BC | Consistency-BC | Diffusion-BC | AWAC | Diffuser | MoRel | Onestep RL | TD3+BC | DT |
|---|---|---|---|---|---|---|---|---|---|
| halfcheetah-m | 42.6 | 31.0±0.4 (46.2±0.4) | 45.4±1.8 (46.3±0.2) | 43.5 | 44.2 | 42.1 | **48.4** | 48.3 | 42.6 |
| hopper-m | 52.9 | 71.7±8.0 (78.3±2.6) | 65.3±5.8 (71.1±5.5) | 57.0 | 58.5 | **95.4** | 59.6 | 59.3 | 67.6 |
| walker2d-m | 75.3 | 83.1±0.3 (84.1±0.3) | 81.2±1.7 (84.3±0.5) | 72.4 | 79.7 | 77.8 | 81.8 | **83.7** | 74.0 |
| halfcheetah-mr | 36.6 | 34.4±5.3 (45.4±0.7) | 41.7±0.4 (44.1±0.3) | 40.5 | **42.2** | 40.2 | 38.1 | 44.6 | 36.6 |
| hopper-mr | 18.1 | **99.7**±0.5 (100.4±0.6) | 67.9±28.1 (99.1±2.3) | 37.2 | 96.8 | 93.6 | 97.5 | 60.9 | 82.7 |
| walker2d-mr | 26.0 | 73.3±5.7 (80.8±2.4) | 77.5±4.7 (80.8±4.5) | 27.0 | 61.2 | 49.8 | 49.5 | **81.8** | 66.6 |
| halfcheetah-me | 55.2 | 32.7±1.2 (39.6±3.4) | 90.8±1.1 (93.5±0.4) | 42.8 | 79.8 | 53.3 | **93.4** | 90.7 | 86.8 |
| hopper-me | 52.5 | 90.6±9.3 (96.8±4.6) | 107.6±4.3 (111.7±0.3) | 55.8 | 107.2 | **108.7** | 103.3 | 98.0 | 107.6 |
| walker2d-me | 107.5 | 110.4±0.7 (111.6±0.7) | 108.9±0.6 (110.5±0.5) | 74.5 | 108.4 | 95.6 | **113.0** | 110.1 | 108.1 |
| **Average** | 51.9 | 69.7 (75.9) | **76.3** (82.4) | 50.1 | 75.3 | 72.9 | 76.1 | 75.3 | 74.7 |

| AntMaze Tasks | BC | Consistency-BC | Diffusion-BC | AWAC | BCQ | BEAR | Onestep RL | TD3+BC | DT |
|---|---|---|---|---|---|---|---|---|---|
| antmaze-u | 54.6 | 75.8±4.0 (87.0±4.5) | 71.8±8.2 (76.8±3.9) | 56.7 | **78.9** | 73.0 | 64.3 | 78.6 | 59.2 |
| antmaze-ud | 45.6 | **77.6**±6.3 (82.4±3.4) | 61.2±9.4 (78.8±7.0) | 49.3 | 55.0 | 61.0 | 60.7 | 71.4 | 53.0 |
| antmaze-mp | 0.0 | **56.8**±30.1 (71.6±14.5) | 43.4±37.8 (56.8±34.5) | 0.0 | 0.0 | 0.0 | 0.3 | 10.6 | 0.0 |
| antmaze-md | 0.0 | **31.6**±22.4 (66.0±6.5) | 29.8±36.3 (69.4±12.3) | 0.7 | 0.0 | 8.0 | 0.0 | 3.0 | 0.0 |
| antmaze-lp | 0.0 | 10.2±4.6 (15.0±3.8) | **14.6**±11.2 (22.4±5.8) | 0.0 | 6.7 | 0.0 | 0.0 | 0.2 | 0.0 |
| antmaze-ld | 0.0 | 12.8±8.2 (19.8±4.0) | **26.6**±10.7 (33.0±8.2) | 1.0 | 2.2 | 0.0 | 0.0 | 0.0 | 0.0 |
| **Average** | 16.7 | **44.1** (57.0) | 41.2 (53.3) | 18.0 | 23.8 | 23.7 | 20.9 | 27.3 | 18.7 |

| Adroit Tasks | BC | Consistency-BC | Diffusion-BC | SAC | BCQ | BEAR | BRAC-p | BRAC-v | REM |
|---|---|---|---|---|---|---|---|---|---|
| pen-human-v1 | 25.8 | 52.4±13.7 (63.7±7.4) | 61.1±5.9 (66.7±4.9) | 4.3 | **68.9** | -1.0 | 8.1 | 0.6 | 5.4 |
| pen-cloned-v1 | 38.3 | 33.4±6.0 (51.9±6.6) | **57.6**±9.5 (62.7±6.1) | -0.8 | 44.0 | 26.5 | 1.6 | -2.5 | -1.0 |
| **Average** | 32.1 | 42.9 (57.8) | **59.4** (64.7) | 1.8 | 56.5 | 12.8 | 4.9 | -1.0 | 2.2 |

| Kitchen Tasks | BC | Consistency-BC | Diffusion-BC | SAC | BCQ | BEAR | BRAC-p | BRAC-v | AWR |
|---|---|---|---|---|---|---|---|---|---|
| kitchen-c | 33.8 | 45.2±5.0 (50.9±3.6) | **76.5**±8.9 (87.3±6.8) | 15.0 | 8.1 | 0.0 | 0.0 | 0.0 | 0.0 |
| kitchen-p | 33.8 | 22.6±3.8 (23.8±2.8) | **50.3**±3.0 (52.9±1.6) | 0.0 | 18.9 | 13.1 | 0.0 | 0.0 | 15.4 |
| kitchen-m | 47.5 | 23.5±1.8 (24.3±1.3) | **56.5**±6.6 (64.7±4.6) | 2.5 | 8.1 | 47.2 | 0.0 | 0.0 | 10.6 |
| **Average** | 38.4 | 30.4 (33.0) | **61.1** | 5.8 | 11.7 | 20.1 | 0.0 | 0.0 | 8.7 |

algorithm used for collecting data in D4RL Gym tasks. Detailed hyperparameters for Consistency-BC and Diffusion-BC refer to Appendix C.1.

Results from Tab. 1 show the advantage of using multi-modal policy representation for offline RL even only with the BC method. For reference purpose, the values in the brackets allow for online evaluation to achieve the best model selection from the set of trained models, which serve as the maximal possible values for the standard offline selection without leveraging online evaluation. Compared with vanilla BC using the Gaussian distribution for policies, Consistency-BC with multi-modality outper-

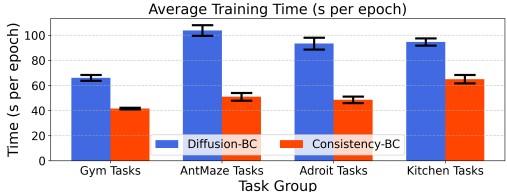

Figure 1: Average training time (seconds per epoch) for Consistency-BC and Diffusion-BC across tasks.

forms it on 14/20 tasks, and Diffusion-BC has better or equivalent performance as BC for 20/20 tasks. Through leveraging multi-modal representation in BC, the improvement of normalized scores averaged over tasks is significant, and this is mainly caused by the multi-modality within the offline dataset by mixing over policies. Moreover, compared with previous offline RL baselines, which do not just apply BC, the Consistency-BC and Diffusion-BC show comparable performances, and even superior performances for tasks like *walker2d-medium-v2*, *hopper-medium-replay-v2*, *walker2d-medium-replay-v2*, *walker2d-medium-expert-v2* in Gym tasks, most of AntMaze, Adroit and Kitchen tasks. The consistency policy is slightly less expressive than the diffusion policy, which is within our expectation due to its heavy reduction on the sampling steps. However, the consistency policy shows higher computational efficiency than diffusion policy as compared in Fig. 1, with an average reduction of **42.97**% computational time across 20 tasks. Detailed computational time for each task is provided in Appendix C.2 Tab. 6.

## 5.2 OFFLINE RL: CONSISTENCY ACTOR-CRITIC

Table 2: The performance of Consistency-AC and SOTA baselines on D4RL Gym, AntMaze, Adroit and Kitchen tasks for offline RL setting. For Consistency-AC and Diffusion-QL, each cell has two values: one for offline model selection and another (in brackets) for online model selection. The bold values are the highest among each row.

| Tasks | CQL | IQL | $\mathcal{X}$-QL | ARQ | IDQL-A | Diffusion-QL | Consistency-AC |
|---|---|---|---|---|---|---|---|
| halfcheetah-m | 44.0 | 47.4 | 48.3 | $45 \pm 0.3$ | 51.0 | $51.1 \pm 0.5\,(51.5 \pm 0.3)$ | $\mathbf{69.1} \pm 0.7\,(71.9 \pm 0.8)$ |
| hopper-m | 58.5 | 66.3 | 74.2 | $61 \pm 0.4$ | 65.4 | $\mathbf{90.5} \pm 4.6\,(96.6 \pm 3.4)$ | $80.7 \pm 10.5\,(99.7 \pm 2.3)$ |
| walker2d-m | 72.5 | 78.3 | 84.2 | $81 \pm 0.7$ | 82.5 | $\mathbf{87.0} \pm 0.9\,(87.3 \pm 0.5)$ | $83.1 \pm 0.3\,(84.1 \pm 0.3)$ |
| halfcheetah-mr | 45.5 | 44.2 | 45.2 | $42 \pm 0.3$ | 45.9 | $47.8 \pm 0.3\,(48.3 \pm 0.2)$ | $\mathbf{58.7} \pm 3.9\,(62.7 \pm 0.6)$ |
| hopper-mr | 95.0 | 94.7 | 100.7 | $81 \pm 24.2$ | 92.1 | $\mathbf{101.3} \pm 0.6\,(102.0 \pm 0.4)$ | $99.7 \pm 0.5\,(100.4 \pm 0.6)$ |
| walker2d-mr | 77.2 | 73.9 | 82.2 | $66 \pm 7.0$ | 85.1 | $\mathbf{95.5} \pm 1.5\,(98.0 \pm 0.5)$ | $79.5 \pm 3.6\,(83.0 \pm 1.5)$ |
| halfcheetah-me | 91.6 | 86.7 | 94.2 | $91 \pm 0.7$ | 95.9 | $\mathbf{96.8} \pm 0.3\,(97.2 \pm 0.4)$ | $84.3 \pm 4.1\,(89.2 \pm 3.3)$ |
| hopper-me | 105.4 | 91.5 | **111.2** | $110 \pm 0.9$ | 108.6 | $111.1 \pm 1.3\,(112.3 \pm 0.8)$ | $100.4 \pm 3.5\,(106.0 \pm 1.3)$ |
| walker2d-me | 108.8 | 109.6 | **112.7** | $109 \pm 0.5$ | **112.7** | $110.1 \pm 0.3\,(111.2 \pm 0.9)$ | $110.4 \pm 0.7\,(111.6 \pm 0.7)$ |
| **Average** | 77.6 | 77.0 | 83.7 | 76.2 | 82.1 | **87.9** (89.3) | 85.1 (89.8) |
| antmaze-u | 74.0 | 87.5 | **93.8** | $97 \pm 0.8$ | 94.0 | $93.4 \pm 3.4\,(96.0 \pm 3.3)$ | $75.8 \pm 1.6\,(85.6 \pm 3.9)$ |
| antmaze-ud | **84.0** | 62.2 | 82.0 | $62 \pm 12.1$ | 80.2 | $66.2 \pm 8.6\,(84.0 \pm 10.1)$ | $77.6 \pm 6.3\,(82.4 \pm 3.4)$ |
| antmaze-mp | 61.2 | 71.2 | 76.0 | $80 \pm 8.3$ | **84.5** | $76.6 \pm 10.8\,(79.8 \pm 8.7)$ | $56.8 \pm 30.1\,(71.6 \pm 14.5)$ |
| **Average** | 73.1 | 73.6 | 83.9 | 79.7 | 86.2 | 78.7 (86.6) | 70.1 (79.9) |
| pen-human-v1 | 35.2 | 71.5 | - | $45 \pm 5.2\,(\text{v0})$ | - | $\mathbf{72.8} \pm 9.6\,(75.7 \pm 9.0)$ | $63.4 \pm 7.7\,(67.9 \pm 5.3)$ |
| pen-cloned-v1 | 27.2 | 37.3 | - | $50 \pm 7.1\,(\text{v0})$ | - | $\mathbf{57.3} \pm 11.9\,(60.8 \pm 11.8)$ | $50.1 \pm 2.2\,(53.7 \pm 3.4)$ |
| **Average** | 31.2 | 54.4 | - | 47.5 | - | **65.1** (68.3) | 56.8 (60.8) |
| kitchen-c | 43.8 | 62.5 | 82.4 | $37 \pm 14.2$ | - | $\mathbf{84.0} \pm 7.4\,(84.5 \pm 6.1)$ | $51.9 \pm 6.0\,(67.6 \pm 2.7)$ |
| kitchen-p | 49.8 | 46.3 | **73.7** | $50 \pm 5.0$ | - | $60.5 \pm 6.9\,(63.7 \pm 5.2)$ | $38.2 \pm 1.8\,(39.8 \pm 1.6)$ |
| kitchen-m | 51.0 | 51.0 | 62.5 | $39 \pm 9.4$ | - | $\mathbf{62.6} \pm 5.1\,(66.6 \pm 3.3)$ | $45.8 \pm 1.5\,(46.7 \pm 0.9)$ |
| **Average** | 48.2 | 53.3 | 72.9 | 42.0 | - | **69.0** (71.6) | 45.3 (51.4) |
| **Total Average** | 66.2 | 69.6 | - | 67.4 | - | **80.3** (83.2) | 72.1 (77.9) |

**Empirical finding 2:** *Replacing diffusion model with consistency model in TD3-BC type algorithm for offline RL will lead to speed up of model training and inference, with slightly worse performances while still outperforming some other baselines.*

For offline RL, the proposed method **Consistency-AC** follows Alg. 3 with consistency model for policy representation, and the consistency policy is embedded in an actor-critic algorithm with BC policy regularization to avoid generating out-of-distribution actions. Results for previous baselines, including CQL (Kumar et al., 2020), IQL (Kostrikov et al., 2021), $\mathcal{X}$-QL (Garg et al., 2023), ARQ (Goo & Niekum, 2022), IDQL-A (Hansen-Estruch et al., 2023) and Diffusion-QL (Wang et al., 2022) are adopted from results reported in corresponding papers. Detailed hyperparameters for Consistency-AC and Diffusion-QL refer to Appendix C.1.

Results from Tab. 2 show the average normalized scores of different methods across five random seeds, with standard deviations reported for Diffusion-QL and Consistency-AC. The results in Tab. 2 are directly comparable with the results in Tab. 1 since they follow the same offline RL setting.

Tab. 2 shows that although Consistency-AC achieves a slightly lower average score (72.1) than Diffusion-QL (80.3), it outperforms the other baselines in most of the tasks, like Gym and Adroit. The AntMaze tasks are found to be hard for the Consistency-AC method, we conjecture that this is potentially caused by the sparse reward signals (as evidenced by Appendix A Fig. 5) in dataset, which makes the difficulty of Q-learning become more of a bottleneck than modeling the multi-modal distributions with behavior cloning. The conservativeness of the $Q$-value estimation might be important but orthogonal to the proposed consistency policy. Considering the reduction of denoising steps in the training and inference stages of Consistency-AC, it can be regarded as a trade-off between the computational efficiency and the approximation accuracy of multi-modal distribution, which will be discussed as following.

| Method | $N$ | Training Time (s per epoch) | Inference Time (ms per sample) | Avg. Norm Score |
|---|---|---|---|---|
| Diffusion-QL | 50 | 206.44±16.70 | 30.65±2.10 | - |
| | 20 | 108.65±2.85 | 13.04±0.90 | 109.2±1.1 (111.1±1.9) |
| | 10 | 76.54±10.74 | 6.87±0.55 | 108.6±0.6 (112.5±0.2) |
| | 5 | 57.06±19.16 | 3.76±0.29 | 108.2±5.6 (112.3±0.2) |
| | 2 | 31.59±10.32 | 1.96±0.10 | 53.6±16.6 (103.5±10.0) |
| | 1 | 30.23±8.75 | 1.37±0.09 | 2.8±1.5 (13.1±12.5) |
| Consistency-AC | 50 | 150.84±31.02 | 26.50±1.92 | - |
| | 20 | 76.22±9.92 | 11.12±0.77 | 101.3±6.3 (107.3±0.2) |
| | 10 | 54.04±4.43 | 5.95±0.44 | 98.4±4.3 (107.1±4.0) |
| | 5 | 40.79±2.79 | 3.39±0.29 | 101.4±4.7 (110.1±1.6) |
| | 2 | 31.94±1.55 | 1.84±0.21 | 102.4±3.0 (106.2±1.6) |
| | 1 | 28.51±1.78 | 1.23±0.11 | 6.2±5.4 (19.1±9.3) |

Table 3: Comparison of computational time for two methods with different denoising steps $N$ on the task *hopper-medium-expert-v2*. The gray lines apply default $N$ values for two models.

**Computational Time.** To evaluate the computational efficiency of Consistency-AC and Diffusion-QL with different denoising steps $N$, we conduct experiments for evaluating the training and inference time for $N \in \{1,2,5,10,20,50\}$ on the *hopper-medium-expert-v2* environment. As generative models based on probability flow, both the consistency model and the diffusion model require the computational time directly dependent on the number of denoising steps $N$, and consistency model (Song et al., 2023) by design requires a smaller number of steps for achieving similar generative performances as the diffusion model. The results are summarized in Tab. 3 for both the training time (seconds per epoch) and the inference time

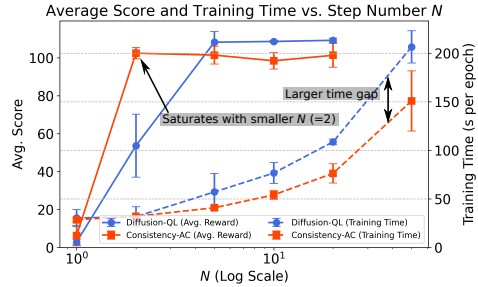

Figure 2: The average normalized scores and training time versus $N$ for two models on *hopper-medium-expert*.

(milliseconds per sample) with the model training using different denoising steps $N$, as well as the average normalized scores for models trained after 2000 epochs with each $N$. Each cell contains the mean and standard deviation over five random seeds. Consistency-AC saturates its performance with only $N=2$ while Diffusion-QL saturates at $N=5$, which consumes about $1.786\times$ more training time

while yielding a slightly better performance ($1.057\times$). The "-" in the table with $N = 50$ indicates a missing value of the average score due to exceeding the limited time (72 hours) for the job. Moreover, as shown in Fig. 2, Consistency-AC has better scaling laws than Diffusion-QL for both training and inference in time consumption with increasing $N$, which is further testified by the linear fitting results in Appendix. C.2 Fig. 6.

**Ablation Studies.** Hansen-Estruch et al. (2023) proposes to use residual networks with layer normalization for network parameterization in diffusion policy, namely LN-Resnet, which is also tested for consistency policy in our experiments. As an ablation study, we compare different variants of Consistency-AC for offline RL setting, including (1) Consistency-BC by setting $\eta = 0$ and without using loss scaling ($\lambda(\tau_n) \equiv 1$ in Eq. 2); (2) only without loss scaling; (3) the standard setting with multi-layer perceptrons (MLP) networks for the parameterization of $f_\theta$; (4) the LN-Resnet parameterization of $f_\theta$. These variants can be regarded as various hyperparameters or training settings for the proposed Consistency-AC algorithm, and the reported results in Tab. 2 are the best choices among these variants. The comparison results for four variants across four task domains are summarized in Fig. 3. Detailed results for this ablation study are shown in Appendix C.3. We find that LN-Resnet does not consistently improve over MLP across tasks for the consistency model but benefits mainly for the Adroit tasks. Without loss scaling, the performance degrades significantly (by $37.8\%$ on average) for most tasks, although for some specific tasks (*e.g.*, AntMaze) it may improve the performance a bit without loss scaling. For most tasks except for AntMaze, Consistency-BC cannot achieve the best performances and the Q-learning loss $\mathcal{L}_q(\theta)$ (as Eq. 5) with proper scaling helps to further improve the scores.

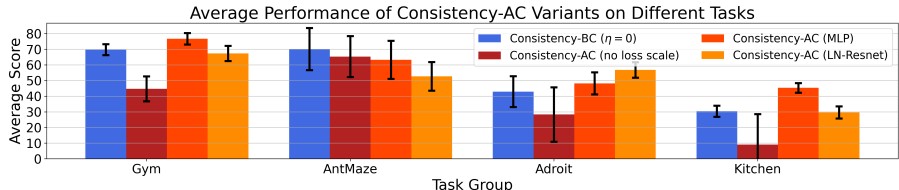

Figure 3: Comparison of variants of Consistency-AC across tasks in offline RL setting.

## 5.3 Offline-to-Online and Online RL

**Empirical finding 3:** *Consistency policy has a close but slightly worse performance than diffusion policy for offline-to-online RL, but a significant improvement of computational efficiency.*

For online RL, we consider both online learning from scratch and the offline-to-online setting with the model trained on offline dataset as an initialization for online fine-tuning. As discussed in previous Sec. 5.2, the offline model can be selected in either an online or offline manner, respectively by model evaluation with or without online experience. Both types of models are used for initializing the policy and value models at the beginning of online fine-tuning. For online fine-tuning, it follows the standard actor-critic algorithm, that the $Q$ value is updated with Eq. 3 using online data, and the policy is updated with the Q-learning loss $\mathcal{L}_q(\theta)$ only as Eq. 5. The algorithms use $\epsilon$-greedy for exploration with a decaying schedule. Pseudo-codes for offline-to-online and online Consistency-AC are provided in Appendix D.1. Hyperparameters for training refer to Appendix D.2.

Table 4: Comparison of normalized scores (last epoch) for methods in offline-to-online and online RL.

| | | | | Offline-to-Online | | | Online | |
|---|---|---|---|---|---|---|---|---|
| **Gym Tasks** | SAC | AWAC | ACA | Diffusion-QL | Consistency-AC | | Diffusion-QL | Consistency-AC |
| halfcheetah-m | 75.2 | 50.5 | 66.6 | **99.6**±2.3 (99.8±1.6) | 98.7±1.8 (97.3±2.9) | | 47.3±2.9 | 55.1±7.0 |
| hopper-m | 73.4 | **97.5** | 96.5 | 77.2±25.6 (60.0±11.8) | 60.5±8.6 (61.8±26.6) | | 82.8±30.6 | 86.3±28.4 |
| walker2d-m | 79.6 | 1.9 | 74.7 | **118.3**±5.8 (117.5±5.9) | 108.9±3.0 (107.9±10.5) | | 77.0±25.7 | 69.4±38.9 |
| halfcheetah-mr | 68.9 | 46.8 | 59.0 | **96.3**±3.9 (97.6±1.2) | 80.7±10.5 (82.3±9.4) | | 43.5±5.7 | 56.5±8.0 |
| hopper-mr | 74.0 | **96.0** | 85.5 | 68.4±20.3 (90.6±24.0) | 74.6±25.1 (63.4±16.7) | | 94.0±12.2 | 75.8±26.8 |
| walker2d-mr | 85.4 | 80.8 | 85.2 | 95.7±18.8 (105.5±13.7) | **102.0**±11.6 (96.5±17.9) | | 87.8±29.0 | 69.0±42.3 |
| halfcheetah-me | 82.2 | 68.8 | 93.7 | **103.9**±2.2 (102.9±1.8) | 99.6±4.1 (95.1±9.7) | | 39.7±3.6 | 56.7±5.8 |
| hopper-me | 65.4 | 73.1 | **98.0** | 71.7±31.1 (67.9±18.6) | 65.4±5.7 (54.7±28.4) | | 62.5±22.2 | 78.6±14.6 |
| walker2d-me | 87.2 | 45.2 | 110.5 | **117.0**±6.3 (111.2±10.6) | 101.8±13.3 (89.2±16.2) | | 74.6±39.0 | 86.2±27.8 |
| **Average** | 76.8 | 62.3 | 85.5 | **94.2** | 88.0 | | 67.7 | 70.4 |

Tab. 4 summarizes the quantitative results for average scores achieved with Consistency-AC and Diffusion-QL across five random seeds for two settings over 9 Gym tasks, as well as offline-to-online baseline methods SAC, AWAC and ACA (Yu & Zhang, 2023). Both the Consistency-AC and Diffusion-QL are pre-trained on the offline dataset for 2000 epochs. Each model is trained for one million steps

for online fine-tuning or learning from scratch. The results for SAC, AWAC and ACA are adopted from the paper (Yu & Zhang, 2023) with each model fine-tuned for 100k online steps. Each cell has two values: one for offline model selection as initialization and another (in brackets) for online model selection as initialization. The normalized scores are slightly lower for Consistency-AC compared with Diffusion-QL in offline-to-online settings, but higher in online RL from scratch. On average, the two methods achieve lower values in online setting than the offline-to-online setting, which testifies the improvement of learning efficiency by initializing with pre-trained generative policy models. However, since the training is set to have a fixed overall timesteps and using the same learning rate $1 \times 10^{-5}$, the purely online models do not converge to its optimal performances yet. The learning rate is chosen for the fine-tuning setting, and the purpose is not to show online RL can achieve scores higher than 100 with sufficient training but to compare with the offline-to-online setting, for demonstrating the effectiveness of initialized models with offline pre-training.

**Empirical finding 4:** *Consistency policy could outperform diffusion policy for online RL setting mostly in computational efficiency and sometimes in sample efficiency, especially for hard tasks.*

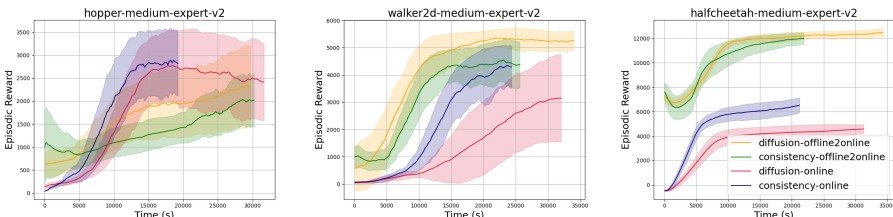

Figure 4: Learning curves of Diffusion-QL and Consistency-AC for online RL and offline-to-online RL with offline model selection in time axis (all trained with one million environment steps). Each curve is smoothed and averaged over five random seeds, and shaded regions show the $95\%$ confidence interval.

Fig. 4 shows the learning curves of Consistency-AC and Diffusion-QL for both offline-to-online and online RL settings with one million online training steps on three example tasks (full results in Appendix D.3 Fig. 8). Different methods consume different time to finish the entire training. The diagrams are plotted with x-axis being the wall-clock time, therefore the curves exhibit different lengths. The diagrams with x-axis being the training steps are shown in Appendix D.4 Fig. 9. The results for offline-to-online setting with online model selection from the offline pre-trained models are provided in Appendix D.4 but with similar performances. For most tasks, the consistency policy has comparable performances with the diffusion policy and a significantly shorter time to finish the entire online training. The offline-to-online methods are usually more sample efficient than the online methods except for three *hopper* tasks, which are relatively easy to learn a good policy. For the online setting, the consistency policies demonstrate significantly more efficient learning than the diffusion policies, especially for more complex tasks like *halfcheetah*. Consistency policies show a sharper score-increasing slope for $8/9$ tasks than the diffusion policies. Our conjecture is that the expressiveness of a model is more essential in offline setting than online setting. For a deterministic optimal policy in MDP, overly expressive policy models like diffusion may hinder the convergence in online setting by being too explorative. For offline-to-online setting, this advantage is less obvious presumably due to the lower initial performances of consistency policies from the offline pre-training. We refer to Appendix D.3 Tab. 9 and Fig. 7 for more analysis of the training time for two methods.

## 6 CONCLUSION

The consistency model as a RL policy strikes a balance between the computational efficiency and the modeling accuracy of multi-modal distribution on offline RL dataset, and achieves comparable performances with the diffusion policies but significant speedup in three typical RL settings. The proposed Consistency-AC algorithm leverages a novel policy representation with policy regularization for offline RL, orthogonal to other offline RL techniques. Future directions include combining the consistency policy with other techniques like conservative $Q$-value estimation for offline RL, better alignment and initialization from offline to online fine-tuning, advanced exploration methods for online RL, etc. Scaling up the task complexity, where more sampling steps are required for the generative models, will reveal a greater potential for consistency policy to show its benefits in retaining expressiveness while reducing the computational cost.

ACKNOWLEDGMENTS

This work was supported by National Science Foundation Grant NSF-IIS-2107304. The authors thank Qinqing Zheng for insightful discussions, and anonymous reviewers for feedback on an early version of this paper.

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

## A  D4RL Dataset Visualization

**T-SNE Plot**

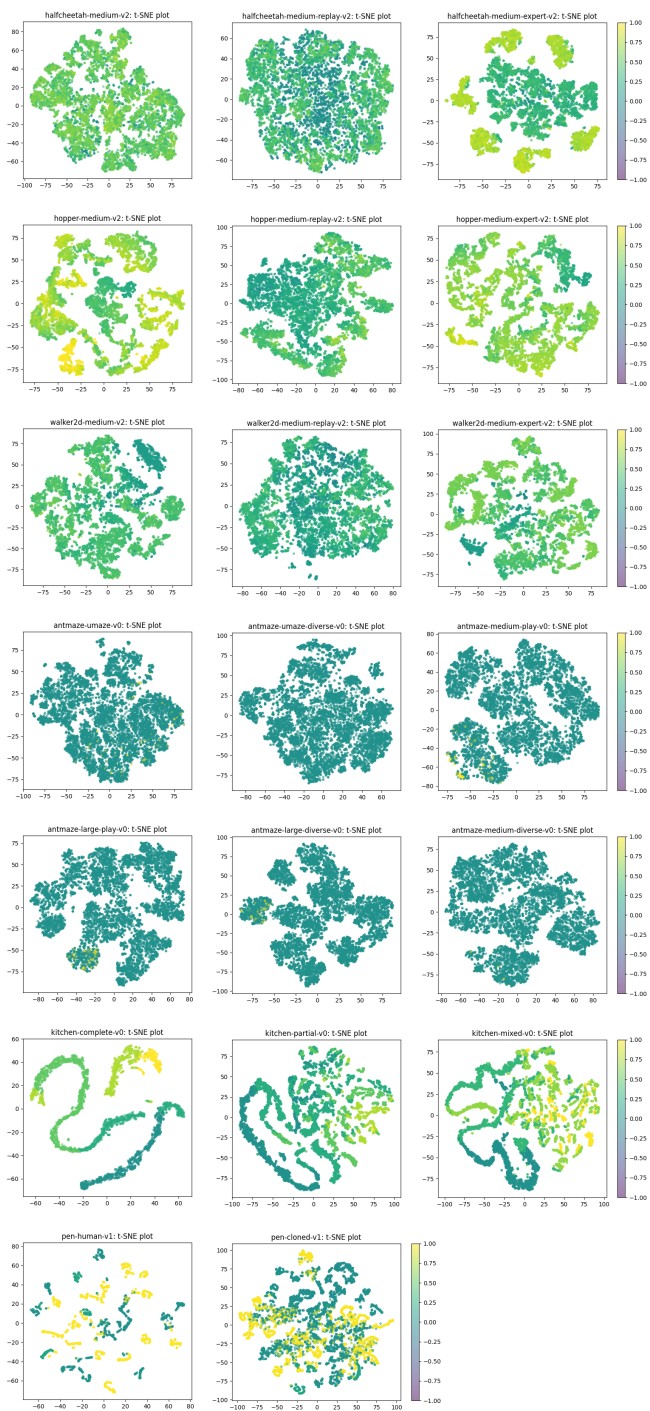

Figure 5: Visualization of t-SNE plots for 10000 (3000 for *pen-human-v1* and *kitchen-complete-v0*) randomly selected $(s,a)$ samples in D4RL dataset, colored by normalized reward (range $[-1,1]$).

## B  CONSISTENCY MODEL TRAINING AND INFERENCE DETAILS

**Training.**  The consistency model $f_\theta$ for modeling data distribution $p_{\text{data}}(\mathbf{x})$ has the loss function (Song et al., 2023):

$$\mathcal{L}_c(\theta) = \mathbb{E}_{n \sim \mathcal{U}(1, N-1), \mathbf{x} \sim p_{\text{data}}(\mathbf{x}), \mathbf{z} \sim \mathcal{N}(\mathbf{0}, \mathbf{I})} \Big[ \lambda(\tau_n) d\big( f_\theta(\mathbf{x} + \tau_{n+1}\mathbf{z}, \tau_{n+1}), f_{\theta^{\top}}(\mathbf{x} + \tau_n \mathbf{z}, \tau_n) \big) \Big] \tag{6}$$

where $d(\cdot, \cdot)$ is the distance metric and we use $l_2$ distance $d(\mathbf{x}, \mathbf{y}) = \|\mathbf{x} - \mathbf{y}\|_2^2$. For training, the sub-sequence $\{\tau_n | n \in [N]\}$ is different from inference, and it follows the Karras boundary (Karras et al., 2022) schedule: $\tau_n = \big( \epsilon^{1/\rho} + \frac{n-1}{N-1}(\tau_N^{1/\rho} - \epsilon^{1/\rho}) \big)^\rho$. The schedule function $N(k) = \lceil \sqrt{\frac{k}{K}((s_1+1)^2 - s_0^2) + s_0^2} - 1 \rceil + 1$ with $k$ as the current training iteration of a total $K$ iterations within one epoch[2].

**Inference.**  After training, the consistency model $f_\theta$ can be used for generating samples given initial noisy input $\hat{\mathbf{x}}_T \sim \mathcal{N}(\mathbf{0}, T^2 \mathbf{I})$, following either single-step sampling $\mathbf{x} = f_\theta(\hat{\mathbf{x}}_T, T)$, or multistep sampling by iteratively calculating $\mathbf{x} = f_\theta(\hat{\mathbf{x}}_{\tau_n}, \tau_n)$ with $\hat{\mathbf{x}}_{\tau_n} = \mathbf{x} + \sqrt{\tau_n^2 - \epsilon^2} \mathbf{z}$ following a given time sequence $\{\tau_n | n \in [N]\}$. For inference, the time sequence is a linspace of $[\epsilon, T]$ with $(N-1)$ sub-intervals as: $\tau_n = \frac{n-1}{N-1}(T - \epsilon) + \epsilon, n \in [N]$.

## C  OFFLINE RL EXPERIMENT DETAILS

### C.1  HYPERPARAMETERS

The offline training of Consistency-BC and Consistency-AC uses a batch size of 256 for training 1000 epochs (500 for *pen-cloned-v1*, 1500 for Kitchen tasks, 2000 for Gym tasks) on D4RL tasks, cosine annealing decaying schedule for learning rates, with other hyperparameters listed in Tab. 5. $\xi = 100.0$ (in $\lambda(\cdot)$) in our experiments for loss scaling in Consistency-AC. Max Q backup (Kumar et al., 2020) is optional. Q norm indicates the normalization of subtracting the mean and dividing the standard deviation for the target Q values in stabilized training. Gradient norm is to clip the $l_2$-norm of the gradients. The Diffusion-BC and Diffusion-QL training follows the hyperparameters of original paper (Wang et al., 2022).

Table 5: The hyperparameters for Consistency-AC in offline (including BC) training on D4RL Gym, AntMaze, Adroit and Kitchen tasks.

| Tasks | Hyperparameters | | | | |
|---|---|---|---|---|---|
| | learning rate | $\eta$ | Q norm | max Q backup | gradient norm |
| halfcheetah-medium-v2 | $3 \times 10^{-4}$ | 1.0 | False | False | 9.0 |
| hopper-medium-v2 | $3 \times 10^{-4}$ | 0.1 | False | False | 9.0 |
| walker2d-medium-v2 | $3 \times 10^{-4}$ | 1.0 | True | False | 1.0 |
| halfcheetah-medium-replay-v2 | $3 \times 10^{-4}$ | 1.0 | False | False | 2.0 |
| hopper-medium-replay-v2 | $3 \times 10^{-4}$ | 0.1 | False | False | 4.0 |
| walker2d-medium-replay-v2 | $3 \times 10^{-4}$ | 0.1 | False | False | 4.0 |
| halfcheetah-medium-expert-v2 | $3 \times 10^{-4}$ | 1.0 | False | False | 7.0 |
| hopper-medium-expert-v2 | $3 \times 10^{-4}$ | 1.0 | False | False | 5.0 |
| walker2d-medium-expert-v2 | $3 \times 10^{-4}$ | 1.0 | True | False | 5.0 |
| antmaze-umaze-v0 | $3 \times 10^{-4}$ | 0.01 | True | False | 2.0 |
| antmaze-umaze-diverse-v0 | $3 \times 10^{-4}$ | 0.01 | True | True | 3.0 |
| antmaze-medium-play-v0 | $1 \times 10^{-3}$ | 0.01 | False | True | 2.0 |
| pen-human-v1 | $3 \times 10^{-5}$ | 0.01 | True | False | 7.0 |
| pen-cloned-v1 | $3 \times 10^{-5}$ | 0.01 | True | False | 8.0 |
| kitchen-complete-v0 | $3 \times 10^{-4}$ | 0.5 | True | False | 2.0 |
| kitchen-partial-v0 | $3 \times 10^{-4}$ | 0.5 | True | False | 2.0 |
| kitchen-mixed-v0 | $3 \times 10^{-4}$ | 0.5 | True | False | 2.0 |

---

[2]Our experiments use constants $\epsilon = 0.002, T = 80; \rho = 7; s_0 = 2, s_1 = 150$ following Song et al. (2023)

## C.2   COMPUTATIONAL TIME

**Overall Training Time.**   Tab. 6 shows the comparison of Diffusion-BC and Consistency-BC in terms of the computational time during training for D4RL Gym, AntMaze, Adroit and Kitchen tasks. Each result is averaged over five random seeds with standard deviations reported. Since different environments are trained for various numbers of total epochs, the comparison is based on per-epoch time consumption. The two methods use the same batch size and number of iterations within each epoch, as well as the same network architecture.

Table 6: The training time (seconds per epoch) for two BC methods on D4RL Gym, AntMaze, Adroit and Kitchen tasks.

| Tasks | Diffusion-BC | Consistency-BC |
|---|---|---|
| halfcheetah-m | 67.93±2.00 | 43.07±0.61 |
| hopper-m | 61.00±1.58 | 38.00±0.49 |
| walker2d-m | 68.17±1.51 | 43.58±0.90 |
| halfcheetah-mr | 67.07±2.07 | 42.75±0.83 |
| hopper-mr | 64.89±3.29 | 38.03±0.47 |
| walker2d-mr | 66.11±1.69 | 42.70±0.54 |
| halfcheetah-me | 67.73±1.78 | 43.60±0.86 |
| hopper-me | 63.04±4.25 | 38.56±0.56 |
| walker2d-me | 69.10±2.83 | 43.88±0.72 |
| **Average** | 66.12±2.33 | 41.57±0.66 |
| antmaze-u | 97.13±4.21 | 47.88±2.59 |
| antmaze-ud | 104.83±4.50 | 47.20±2.23 |
| antmaze-mp | 109.66±3.82 | 57.92±4.46 |
| antmaze-md | 112.25±2.41 | 51.80±2.59 |
| antmaze-lp | 113.15±1.01 | 56.52±3.17 |
| antmaze-ld | 118.62±3.42 | 53.89±2.74 |
| **Average** | 109.27±3.44 | 52.54±3.05 |
| pen-human-v1 | 92.20±3.17 | 46.94±2.92 |
| pen-cloned-v1 | 94.64±6.16 | 50.33±2.23 |
| **Average** | 93.42±4.67 | 48.64±2.58 |
| kitchen-c | 96.77±2.74 | 66.71±4.64 |
| kitchen-p | 94.25±2.27 | 61.85±2.74 |
| kitchen-m | 93.02±3.49 | 66.60±2.59 |
| **Average** | 94.68±2.83 | 65.05±3.32 |
| **Total Average** | 86.08±3.16 | 49.09±2.34 |

**Scaling Law.**   Fig. 6 further shows the scaling laws of training time and inference time with increasing $N$ for Diffusion-QL and Consistency-AC in offline RL setting, based on results in Tab. 3 for environment *hopper-medium-expert-v2*. Notice that the coefficients of Consistency-AC are smaller than Diffusion-QL in both training (2.47 vs. 3.54) and inference (0.515 vs. 0.598), which indicates smaller time consumption with increasing $N$.

## C.3   ABLATION STUDIES

Four variants of Consistency-AC are compared for offline RL setting, including (1) Consistency-BC by setting $\eta = 0$ and without using loss scaling ($\lambda(\tau_n) \equiv 1$ in Eq. 2); (2) only without loss scaling; (3) the standard setting with MLP networks for the parameterization of $f_\theta$; (4) the LN-Resnet parameterization of $f_\theta$. These variants can be regarded as various hyperparameters or training settings for the proposed

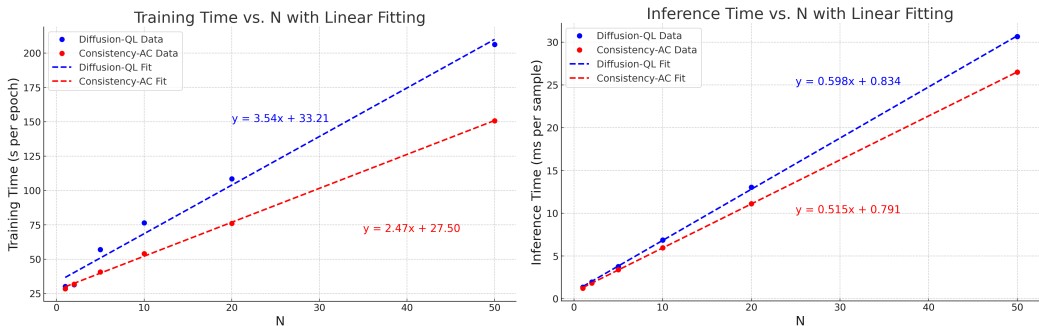

Figure 6: The training time (left) and inference time (right) versus denoising steps $N$ for Diffusion-QL and Consistency-AC in offline RL, evaluated on *hopper-medium-expert-v2* environment.

Consistency-AC algorithm. The average scores for five random seeds over D4RL Gym, AntMaze, Adroit and Kitchen are shown in Tab. 7.

Table 7: The performance of Consistency-AC variants on D4RL Gym, AntMaze, Adroit and Kitchen tasks for offline RL setting. Each cell has two values: one for offline model selection and another (in brackets) for online model selection. Each result is averaged over five random seeds with standard deviations reported.

| Gym Tasks | Consistency-BC ($\eta=0$) | Consistency-AC (no loss scale) | Consistency-AC (MLP) | Consistency-AC (LN-Resnet) |
|---|---|---|---|---|
| halfcheetah-m | 31.0±0.4 (46.2±0.4) | 69.1±0.7 (71.9±0.8) | 50.1±0.4 (50.4±0.2) | 50.6±0.3 (50.9±0.2) |
| hopper-m | 71.7±8.0 (78.3±2.6) | 80.7±10.5 (99.7±2.3) | 78.0±3.9 (86.4±4.0) | 74.1±6.7 (83.7±8.5) |
| walker2d-m | 83.1±0.3 (84.1±0.3) | 5.5±1.7 (21.2±1.6) | 63.0±5.2 (75.0±1.8) | 66.2±5.2 (75.4±1.9) |
| halfcheetah-mr | 34.4±5.3 (45.4±0.7) | 58.7±3.9 (62.7±0.6) | 47.3±0.2 (47.8±0.3) | 47.8±0.3 (48.4±0.1) |
| hopper-mr | 99.7±0.5 (100.4±0.6) | 80.2±9.0 (103.4±1.2) | 94.5±6.4 (100.9±0.2) | 98.7±2.9 (100.6±0.3) |
| walker2d-mr | 73.3±5.7 (80.8±2.4) | 72.3±15.4 (105.1±1.6) | 76.8±5.5 (86.1±1.2) | 79.5±3.6 (83.0±1.5) |
| halfcheetah-me | 32.7±1.2 (39.6±3.4) | 22.6±10.4 (55.2±11.6) | 84.3±4.1 (89.2±3.3) | 61.7±13.6 (68.4±6.7) |
| hopper-me | 90.6±9.3 (96.8±4.6) | 10.1±16.2 (10.8±15.6) | 100.4±3.5 (106.0±1.3) | 43.1±5.7 (54.5±11.2) |
| walker2d-me | 110.4±0.7 (111.6±0.7) | 2.7±4.3 (14.9±6.9) | 91.1±3.4 (97.7±3.2) | 84.1±5.1 (97.5±1.6) |
| **Average** | 69.7 (75.9) | 44.7 (60.5) | 76.7 (82.2) | 67.3 (73.6) |
| **AntMaze Tasks** | **Consistency-BC ($\eta=0$)** | **Consistency-AC (no loss scale)** | **Consistency-AC (MLP)** | **Consistency-AC (LN-Resnet)** |
| antmaze-u | 75.8±4.0 (87.0±4.5) | 75.4±5.8 (82.6±3.8) | 68.8±2.3 (82.2±4.7) | 75.8±1.6 (85.6±3.9) |
| antmaze-ud | 77.6±6.3 (82.4±3.4) | 75.2±6.6 (80.2±2.8) | 68.6±4.4 (78.4±1.1) | 72.4±3.5 (81.2±1.9) |
| antmaze-mp | 56.8±30.1 (71.6±14.5) | 45.2±26.9 (73.2±8.4) | 52.2±29.8 (70.4±7.1) | 10.0±22.4 (59.4±12.8) |
| **Average** | 70.1 (80.3) | 65.3 (78.7) | 63.2 (77.0) | 52.7 (75.4) |
| **Adroit Tasks** | **Consistency-BC ($\eta=0$)** | **Consistency-AC (no loss scale)** | **Consistency-AC (MLP)** | **Consistency-AC (LN-Resnet)** |
| pen-human-v1 | 52.4±13.7 (63.7±7.4) | 8.4±24.0 (22.1±20.5) | 60.6±10.2 (66.6±7.5) | 63.4±7.7 (67.9±5.3) |
| pen-cloned-v1 | 33.4±6.0 (51.9±6.6) | 48.2±10.8 (58.2±12.6) | 35.8±3.9 (40.5±2.6) | 50.1±2.2 (53.7±3.4) |
| **Average** | 42.9 (57.8) | 28.3 (40.2) | 48.2 (53.6) | 56.8 (60.8) |
| **Kitchen Tasks** | **Consistency-BC ($\eta=0$)** | **Consistency-AC (no loss scale)** | **Consistency-AC (MLP)** | **Consistency-AC (LN-Resnet)** |
| kitchen-c | 45.2±5.0 (50.9±3.6) | 10.0±20.1 (25.5±24.6) | 51.9±6.0 (67.6±2.7) | 36.9±3.2 (38.0±2.5) |
| kitchen-p | 22.6±3.8 (23.8±2.8) | 7.7±16.9 (17.0±14.5) | 38.2±1.8 (39.8±1.6) | 25.8±5.5 (28.6±2.7) |
| kitchen-m | 23.5±1.8 (24.3±1.3) | 9.7±21.3 (15.8±20.2) | 45.8±1.5 (46.7±0.9) | 26.0±3.0 (28.8±2.1) |
| **Average** | 30.4 (33.0) | 9.1 (19.4) | 45.3 (51.4) | 29.6 (31.8) |
| **Total Average** | 59.7 (67.0) | 40.1 (54.1) | 64.5 (72.5) | 56.8 (65.0) |

# D OFFLINE-TO-ONLINE AND ONLINE RL DETAILS

## D.1 ALGORITHMS

---

**Algorithm 4** Offline-to-Online Consistency Actor-Critic

**Input** offline pretrained policy $\pi_\theta$ and critic networks $Q_{\phi_1}, Q_{\phi_2}$

Initialize online dataset $\tilde{\mathcal{D}} = \emptyset$, target network parameters: $\theta^\mathsf{T} \leftarrow \theta$, $\phi_1^\mathsf{T} \leftarrow \phi_1, \phi_2^\mathsf{T} \leftarrow \phi_2$

**for** episode $j = 1, ..., M$ **do**

    Reset the environment and observe $s_1$.

    **for** $t = 1, ..., H$ **do**

        % Collect Samples

        Infer action $a_t$ based on $s_t$ with consistency policy $\pi_\theta$ by Alg. 1.

        Execute actions $a_t$, observe reward $r_t$, next state $s_{t+1}$.

        Store data sample $(s_t, a_t, r_t, s_{t+1})$ into $\tilde{\mathcal{D}}$.

        Sample minibatch $\mathcal{B} = \{(s, a, r, s')\} \subseteq \tilde{\mathcal{D}}$;

        % Q-value Update

        Update $Q_{\phi_1}, Q_{\phi_2}$ with Eq. 3;

        % Policy Update

        Update policy $\pi_\theta$ (with model $f_\theta$) via loss $\mathcal{L}_q(\theta)$ as Eq. 5;

        % Target Update

        Update target: $\theta^\mathsf{T} \leftarrow \alpha\theta^\mathsf{T} + (1-\alpha)\theta, \phi_i^\mathsf{T} \leftarrow \alpha\phi_i^\mathsf{T} + (1-\alpha)\phi_i, i \in \{1, 2\}$;

    **end for**

**end for**

---

**Algorithm 5** Online Consistency Actor-Critic

Initialize policy $\pi_\theta$ and critic networks $Q_{\phi_1}, Q_{\phi_2}$

Initialize online dataset $\tilde{\mathcal{D}} = \emptyset$, target network parameters: $\theta^\mathsf{T} \leftarrow \theta$, $\phi_1^\mathsf{T} \leftarrow \phi_1, \phi_2^\mathsf{T} \leftarrow \phi_2$

**for** episode $j = 1, ..., M$ **do**

    Reset the environment and observe $s_1$.

    **for** $t = 1, ..., H$ **do**

        % Collect Samples

        Infer action $a_t$ based on $s_t$ with consistency policy $\pi_\theta$ by Alg. 1.

        Execute actions $a_t$, observe reward $r_t$, next state $s_{t+1}$.

        Store data sample $(s_t, a_t, r_t, s_{t+1})$ into $\tilde{\mathcal{D}}$.

        Sample minibatch $\mathcal{B} = \{(s, a, r, s')\} \subseteq \tilde{\mathcal{D}}$;

        % Q-value Update

        Update $Q_{\phi_1}, Q_{\phi_2}$ with Eq. 3;

        % Policy Update

        Update policy $\pi_\theta$ (with model $f_\theta$) via loss $\mathcal{L}_q(\theta)$ as Eq. 5;

        % Target Update

        Update target: $\theta^\mathsf{T} \leftarrow \alpha\theta^\mathsf{T} + (1-\alpha)\theta, \phi_i^\mathsf{T} \leftarrow \alpha\phi_i^\mathsf{T} + (1-\alpha)\phi_i, i \in \{1, 2\}$;

    **end for**

**end for**

---

## D.2 HYPERPARAMETERS

Table 8: The hyperparameters for Consistency-AC offline-to-online and online training on Gym tasks.

| Hyperparameter | Value |
|---|---|
| learning rate | $1 \times 10^{-5}$ |
| batch size | 256 |
| $\epsilon$-greedy schedule | linear |
| $\epsilon_0$ | 1.0 |
| $\epsilon_\infty$ | 0.01 |
| exploration fraction | 0.1 |
| discount $\gamma$ | 0.99 |
| buffer size | $1 \times 10^5$ |

## D.3 COMPUTATIONAL TIME

The overall training time for one million environment steps using Diffusion-QL and Consistency-AC in offline-to-online and online RL settings is shown in Tab. 9, with the average training time for each setting summarized in Fig. 7. The reduction of computational time in online setting is less significant than the offline setting (as Fig. 1) because there is a large portion of time consumed by the environment simulation steps following the agent's action inference. The improvement of model inference and update will not affect the environment simulation time.

Table 9: The overall training time (hours) for offline-to-online and online settings on Gym tasks.

| | Offline-to-Online | | Online | |
|---|---|---|---|---|
| Gym Tasks | Diffusion-QL | Consistency-AC | Diffusion-QL | Consistency-AC |
| halfcheetah-m | 11.55±4.08 | 9.60±2.33 | 9.09±0.88 | 8.77±0.96 |
| hopper-m | 8.97±1.48 | 6.90±0.79 | 8.06±0.88 | 6.99±0.95 |
| walker2d-m | 9.17±0.29 | 8.19±1.91 | 8.23±1.04 | 6.98±0.89 |
| halfcheetah-mr | 9.18±0.22 | 7.72±0.89 | 8.72±0.88 | 7.76±1.02 |
| hopper-mr | 8.22±0.20 | 7.26±1.69 | 8.12±0.81 | 6.92±0.78 |
| walker2d-mr | 8.85±0.22 | 7.32±0.88 | 8.07±1.01 | 7.05±1.05 |
| halfcheetah-me | 9.24±0.21 | 8.46±1.96 | 8.54±0.74 | 7.65±1.02 |
| hopper-me | 8.28±0.20 | 7.47±0.57 | 8.02±1.07 | 7.01±0.99 |
| walker2d-me | 9.93±1.27 | 9.41±2.26 | 8.35±0.73 | 8.29±0.86 |
| Average | 9.27±0.91 | 8.04±1.48 | 8.36±0.89 | 7.49±0.95 |

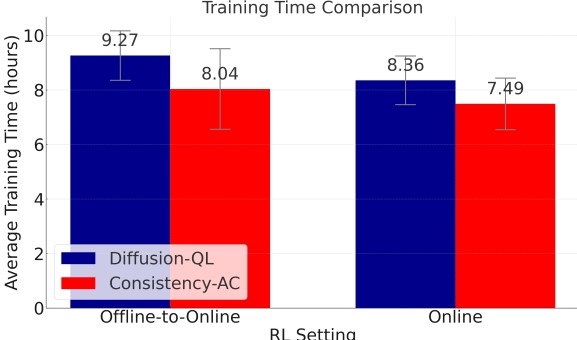

Figure 7: The average training time (hours) for offline-to-online and online training with Diffusion-QL and Consistency-AC on 9 Gym tasks.

## D.4 MORE RESULTS

Table 10: Comparison of scores (unnormalized, maximum over epochs) for methods in offline-to-online and online RL settings.

| | Offline2Online | | Online | |
|---|---|---|---|---|
| Task | Diffusion-QL | Consistency-AC | Diffusion-QL | Consistency-AC |
| halfcheetah-m | **12445.9**±315.8 (12428.4±222.9) | 12280.3±124.1 (12273.4±206.3) | 5745.9±388.5 | 6725.2±944.4 |
| hopper-m | 3626.1±50.9 (3465.9±236.4) | 3595.8±153.6 (3448.1±243.6) | **3673.5**±47.7 | 3589.7±163.4 |
| walker2d-m | **5774.8**±217.3 (5561.4±260.2) | 5536.1±360.3 (5662.0±114.3) | 4316.2±612.1 | 3790.9±1677.5 |
| halfcheetah-mr | **12060.1**±265.7 (12198.6±168.9) | 9941.1±1343.2 (10274.1±1312.7) | 5218.5±726.2 | 6890.3±963.1 |
| hopper-mr | 3657.0±263.3 (3855.4±84.7) | 3262.4±708.1 (3652.2±390.0) | **3663.9**±29.7 | 3418.9±613.2 |
| walker2d-mr | **5240.6**±682.8 (5584.5±347.0) | 5092.3±408.5 (5394.4±708.3) | 4675.4±214.5 | 3918.7±1673.3 |
| halfcheetah-me | **12916.7**±202.5 (12676.7±178.1) | 12480.7±359.3 (11916.9±1156.7) | 4725.6±407.9 | 6889.6±657.5 |
| hopper-me | 3503.5±490.8 (3527.2±297.2) | 3536.9±147.5 (5561.4±260.2) | 3668.8±37.5 | **3701.0**±49.2 |
| walker2d-me | **5630.5**±209.0 (5720.7±377.9) | 5447.0±307.9 (5568.1±553.6) | 3887.1±1701.5 | 5040.0±166.0 |
| Average | **7206.1** | 6797.0 | 4397.2 | **4884.9** |

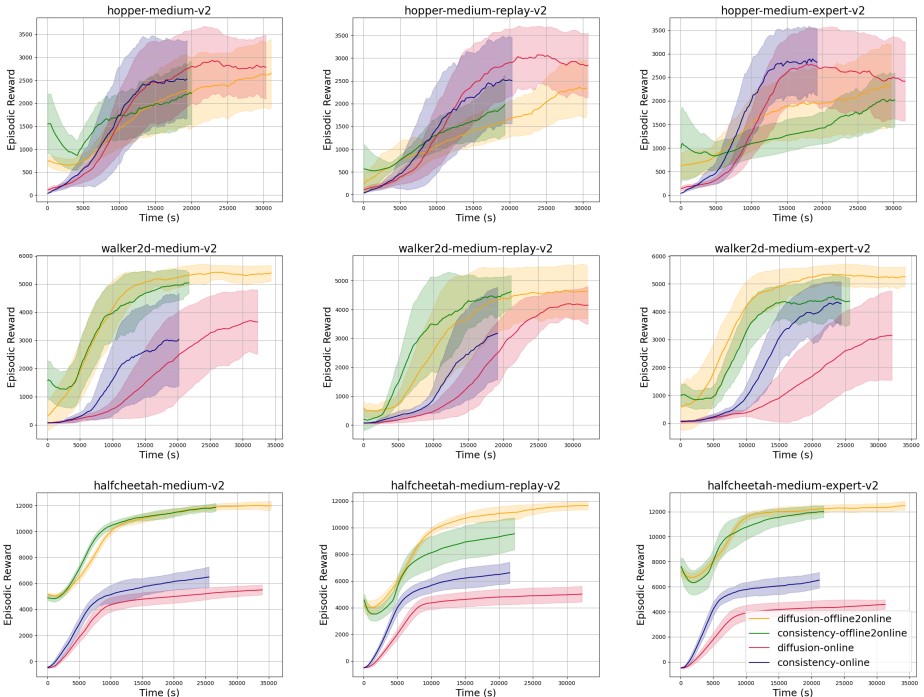

Figure 8: Learning curves of Diffusion-QL and Consistency-AC for online RL and offline-to-online RL with offline model selection in time axis (all trained with one million environment steps). Each curve is smoothed and averaged over five random seeds, and the shaded regions show the $95\%$ confidence interval.

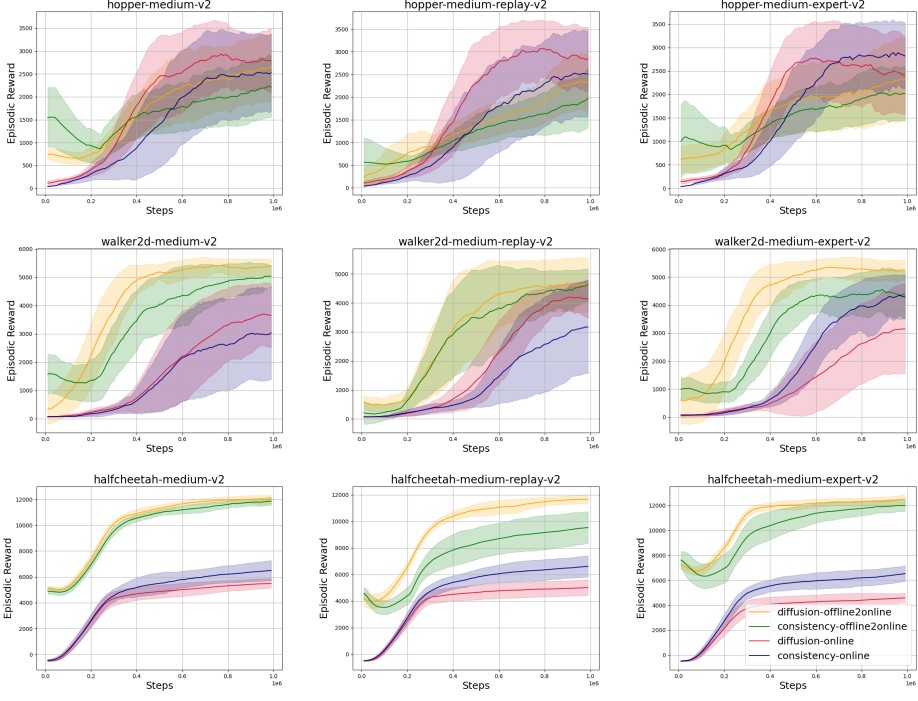

Figure 9: Learning curves of Diffusion-QL and Consistency-AC for online RL and offline-to-online RL with offline model selection in step axis. Each curve is smoothed and averaged over five random seeds, and the shaded regions show the $95\%$ confidence interval.

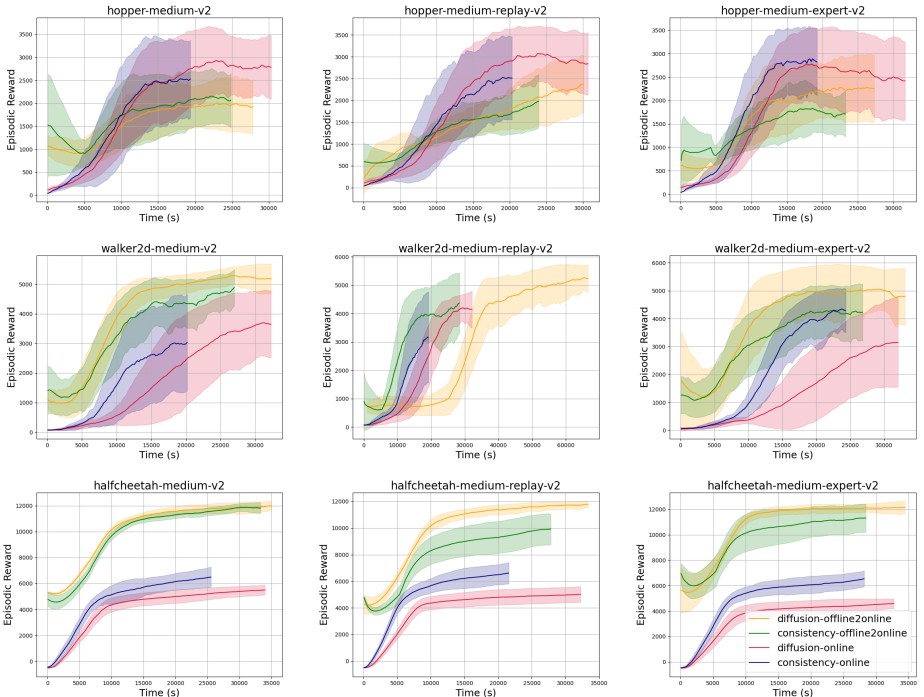

Figure 10: Learning curves of Diffusion-QL and Consistency-AC for online RL and offline-to-online RL with online model selection in time axis (all trained with one million environment steps). Each curve is smoothed and averaged over five random seeds, and the shaded regions show the 95% confidence interval.

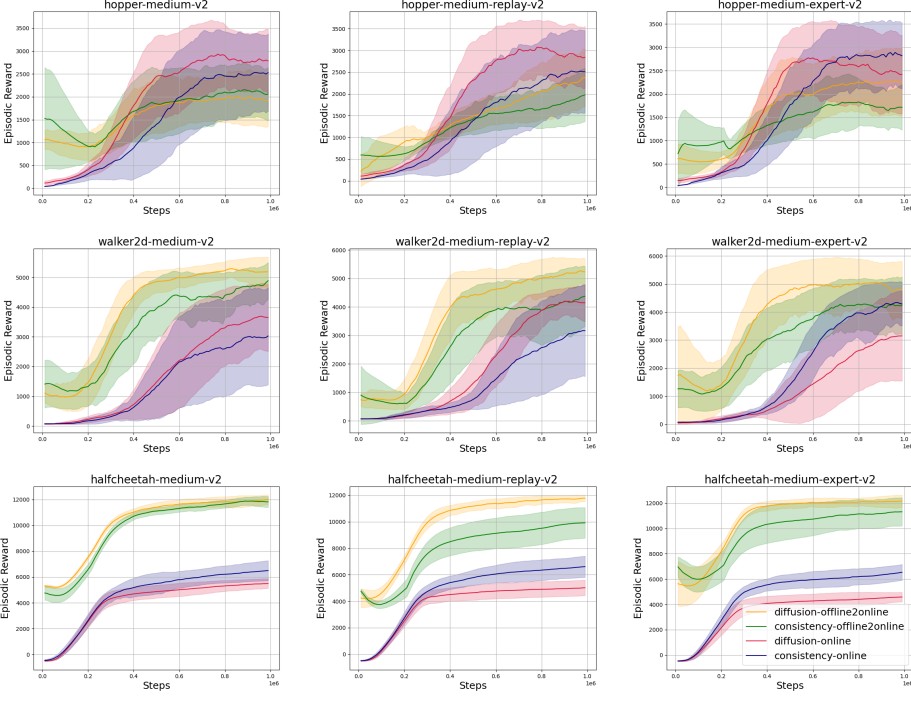

Figure 11: Learning curves of Diffusion-QL and Consistency-AC for online RL and offline-to-online RL with online model selection in step axis. Each curve is smoothed and averaged over five random seeds, and the shaded regions show the 95% confidence interval.

