# OpenReview forum: "Consistency Models as a Rich and Efficient Policy Class for Reinforcement Learning"
_ICLR.cc/2024/Conference — ICLR 2024 poster_

### Official Review · Reviewer_JCUm · 2023-10-27

**Soundness:** 2 fair
**Presentation:** 3 good
**Contribution:** 2 fair
**Rating:** 3
**Confidence:** 4

**Summary:**

It is known that the inference process of the diffusion model can be slow. In the context of RL, the diffusion model has been introduced and widely adopted recently. The authors focus on addressing the slow inference speed issue of diffusion in this paper. Their solution is to replace the diffusion model with the recently proposed consistency model. The authors make some minor adaptations of consistency models to make it fit the RL setting. The authors also test their method across offline RL setting, online RL setting, and offline2online finetuning setting, by building a consistency model on top of behavior cloning and actor-critic structure.

**Strengths:**

# Strengths

This paper pinpoints an interesting question that the current diffusion-based offline RL algorithms (e.g., Diffusion-QL) suffer from slow inference speed. The authors then propose to address this issue by using an existing method, the consistency model, and adapting it to the offline RL setting. This, as far as the reviewer can tell, is the first work that introduces the consistency model into the offline RL setting, online RL setting and offline2online setting. This paper is generally quite well-written, and I enjoy reading this work. The structure of this paper is clear and easy to follow. It is easy for the readers to capture the core points/conclusions in the experiment section. The authors conduct several experiments in D4RL domains like MuJoCo, AntMaze, etc., to show that their method can reduce the training time and inference time while maintaining comparable performance. The authors also do a good job in the related work part.

**Weaknesses:**

# Weaknesses

Despite the aforementioned strengths of this paper, I think this paper is below the acceptance bar of this venue. Please refer to the following comments

- (major) The novelty of this paper is limited. The authors simply borrow an existing method from the vision community and apply it to the RL tasks, with some minor modifications. I do not see much novelty in doing so. This paper seems more like a technical report or experimental report, in that the authors conduct several experiments and summarize the conclusions. One serious flaw of this paper is that the author merely reports some experimental phenomenons while the corresponding explanations and discussion are unfortunately missing. From an ICLR paper of this kind, I would expect to understand why such a phenomenon occurs. This paper leaves me with more questions than answers. For example, why on many offline and offline2online tasks, the consistency model underperform the diffusion model, while on some online hard tasks, the consistency model seems to be better?

- (major) This paper does not consider statistical significance. Written statements and the presentation of the results as tables (often without standard deviations) obscure this flaw. In fact, ALL tables in this paper do not include any signal of statistical significance for baseline methods, e.g., std, IQM [1]. We have reached a point of maturity in the field where claims need to be made in reference to actual statistical evidence, which seems to be lacking in the current presentation.

[1] Deep reinforcement learning at the edge of the statistical precipice. NeurIPS

- (major) The proposed consistency-BC or consistency-AC do not show much improvement over the diffusion-based counterparts. If one looks at the experiments in this paper, it is clear that the consistency-BC and consistency-AC cannot beat diffusion-BC or diffusion-QL on most of the tasks. The authors say that consistency-AC or consistency-BC can achieve less training time. Well, I do not see this as an appealing advantage over the diffusion-based methods, since training cost is somewhat unimportant, while the most critical part, from my perspective, is the inference speed. Let us then take a look at the inference cost. Based on Table 3 in the main text, it is clear that the inference speed of diffusion-QL is quite similar to that of the consistency-AC. For instance, when setting $N=5$, the inference speed of diffusion-QL gives 3.76ms and consistency-AC gives 3.39ms, while their performance differs (diffusion-QL has an average score of 108.2, while consistency-AC only has 101.4). I actually do not see many advantages of utilizing consistency-AC or consistency-BC in practice

- (major) Even worse, it seems that the performance of consistency-AC and consistency-BC are acquired by carefully tuning the hyperparameters. As a piece of evidence, one can see the hyperparameter setup in Appendix C (Table 5). This indicates that the generality and effectiveness of the proposed method are limited

- (minor) On page 3, Eq 3 is the objective using clipped double Q-learning instead of vanilla double Q-learning. The authors ought to cite the TD3 paper here instead of the double Q-learning paper.

- (minor) No codes are provided in this paper, and the authors do not include a reproducibility statement section in the main text

**Questions:**

- why do you use different baselines on different domains? It is somewhat confusing that you use some quite weak baselines on domains like antmaze, adroit, and kitchen. As an example, why do you use AWR, BRAC, and REM on adroit tasks and kitchen tasks in Table 1? It seems to me that the advantages of consistency-BC and consistency-AC are illustrated by carefully picking the baselines.

- how important is the gradient norm to the consistency-BC and consistency-AC algorithms?

---

> ### Author Response · Authors · 2023-11-16
> **Response to review [1/2]**
>
> We appreciate the insightful comments and suggestions from the reviewer. We will try our best to answer those questions raised by the reviewer to remove his/her concerns.
>
> The novelty concern is replied in the above general response.
>
> We appreciate the reviewer for raising the question regarding why consistency models are able to achieve higher online performances than diffusion models. Our conjecture is that the expressiveness of a model is more essential in offline setting than online setting, especially when multi-modal distributions appear in offline dataset and the algorithm is of the behavior-cloning (BC) type. Our results of BC in offline setting as Tab.1 show that the expressive policies can significantly improve the BC performances, which can be helpful for policies to quickly converge in offline RL.  However for online RL in Markov decision process, the theoretically optimal policy can always be deterministic, which indicates that the expressive models will shrink back to unimodal to approach optimality, thus expressiveness may only help with initial exploration instead of final convergence of the policy. This conjecture can be hard to verify consistently due to the heterogeneous distributions of different tasks. However, this is an interesting question and we will explain this in a modified version of the paper.
>
> **Statistical significance**.
> For all the results derived by ourselves, we reported the standard deviations to show its statistical significance, for both algorithm performances as Tab.1,2,4 and Fig. 3 and time consumption as Tab. 3, also the learning curves in Fig. 4 are shaded by 95% confidence intervals. Other results without standard deviations are adapted from previous papers, where most of the values are reported without statistical significance. The results of BC, BEAR, BRAC, AWR, BCQ, SAC and CQL can be found in D4RL paper and CQL paper, all without standard deviations reported. The results of DT, TD3+BC, Onestep RL, AWAC, IQL can be found in IQL paper, Diffuser and IDQL can be found in IDQL paper, results of $\mathcal{X}$-QL and MoRel can be found in their own papers, these are all reported without standard deviations as well. For Tab. 1 and 2, we manage to find the standard deviations reported for ARQ and already fill the values in the modified draft. The results of SAC, AWAC, ACA in Tab. 4 are also reported without standard deviations in ACA paper. It turns out our paper is the very few papers reporting the statistical significance of our own algorithms in the domain. We appreciate the attention of the reviewer for the statistical significance of the results and we have tried our best to satisfy this requirement.
>
> **Performances**.
> We admit the current consistency model does not yield better performances than diffusion models for offline RL settings, and we do not expect it since otherwise the consistency model will not just be an acceleration method of the diffusion model but a total replacement of it. There is always the trade-off of computational efficiency and the model accuracy. We agree that the inference speed is more critical for RL. However, the Tab. 3 may be interpreted in another way by the reviewer as we intended. As also in above general response, in Tab. 3, our major claim is that Consistency-AC can achieve a similar performance as Diffusion-QL with a much smaller $N$, instead of comparing with the same $N$. Consistency-AC saturates its performances with $N=2$ for most of the tasks in our experiments rather than just this one. Specifically, if we look at $N=2$ for Consistency-AC while $N=5$ for Diffusion-QL, the scores are similar but Consistency-AC saves half inference time (1.84/3.76) and training time (31.94/57.06). By default in our experiments, Consistency-AC and Consistency-BC use $N=2$ while Diffusion-BC and Diffusion-QL use $N=5$ for results reported in paper.

---

> ### Author Response · Authors · 2023-11-16
> **Response to review [2/2]**
>
> **Hyperparameter tuning**. We may disagree with the reviewer that hyperparameter tuning is making the proposed method less convincing. Actually, the hyperparameters in Tab. 5 are all inherited from Diffusion-QL paper [3] except for the Q norm, some are not reported in the previous paper but can be found in their experiments. Most of the present state-of-the-art algorithms for specific domains require hyperparameter tuning, which is quite common in literature, with evidence in Tab. 3 of paper [3] and Tab. 4 of paper [4]. They may not report full hyperparameters in the algorithms, and some other papers do not even report their hyperparameters, but this does not indicate that there is no hyperparameter tuning in their reported results. We don’t see any reason that our reported results with good hyperparameter tuning indicates the generality of the method is bad. We can understand the concern if the reviewer is asking how robust the proposed method is against each hyperparameter. This can be computationally heavy to evaluate and we do not see a lot of previous work reporting it. This is further answered in later question regarding the choice of gradient norm.
>
> We thank the reviewer for pointing out the mis-reference of Eq. 3, and will modify it in the modified version.
>
> We plan to open-source the code after the review process and the clean-up of the code. We have added the statement in the modified version, thanks for pointing out.
>
> For the questions of different baseline on different domains, those baseline results are adapted from previous Diffusion-QL paper [3] but not cherry-picked by us. We guess one reason that those baselines are filled on *Adroit* and *Kitchen* tasks is because some strong baselines (like MoRel, DT, Diffuser, etc) on Gym tasks do not report their values on *Adroit* and *Kitchen* tasks while these 'weak' baselines are reported. Our paper just tries to report thoroughly about previous baselines but not take any cherry-picking.
>
> For the question regarding the gradient norm, most of the gradient norm values are directly inherited from Diffusion-QL paper [3] given the usage of the same neural network architectures, which means we do not thoroughly search over the values. Exception exists for *Kitchen* tasks where we reduce the gradient norm values to stabilize training. However, this is still a rough tuning instead of a thorough searching over the values. A thorough search over hyperparameters can be computationally very expensive. We found the Q value normalization and coefficient $\eta$ to be more influential on performances than gradient norm.
>
> References:
>
> [3] Wang, Zhendong, Jonathan J. Hunt, and Mingyuan Zhou. "Diffusion policies as an expressive policy class for offline reinforcement learning." arXiv preprint arXiv:2208.06193 (2022).
>
> [4] Garg, Divyansh, et al. "Extreme q-learning: Maxent RL without entropy." arXiv preprint arXiv:2301.02328 (2023).

---

> > ### Comment · Reviewer_JCUm · 2023-11-22
> >
> > Thanks for your rebuttal. Please find the comments below.
> >
> > > novelty
> >
> > I hold my opinion that simply borrowing an existing method from the vision community and applying it to the RL tasks is under the bar of top-tier conferences like ICLR. However, I agree with the authors that it is valuable to propose the usage of more efficient yet expressive models for RL, and agree that there are some modifications to the consistency model. I would not criticize the authors too much on this point.
> >
> > > why on many offline and offline2online tasks, the consistency model underperform the diffusion model, while on some online hard tasks, the consistency model seem to be better?
> >
> > The authors do not explain this well. Offline2online tasks also involve the online phase, while the consistency model can underperform the diffusion model, while on some online tasks from scratch, the consistency model can be better. The inherent reasons are not clear. I would say further experiments are needed. I understand that the rebuttal period of this venue is about to close, and I hope the authors can add these to future versions of this manuscript.
> >
> > > It turns out our paper is the very few papers reporting the statistical significance of our own algorithms in the domain
> >
> > Cannot agree with that. Numerous existing papers consider statistical significance, e.g., [1,2]. I strongly believe in the realm of RL, it is vital to consider the statistical significance of both baselines and the proposed methods. The authors can find baseline results with statistical significance in other published papers.
> >
> > [1] Uncertainty-Based Offline Reinforcement Learning with Diversified Q-Ensemble
> >
> > [2] Revisiting the Minimalist Approach to Offline Reinforcement Learning
> >
> > > performance
> >
> > Thanks for the clarification. But I do not think my previous comments are *interpreted in another way*. I can now see that the consistency model can aid faster convergence, while it still underperforms the diffusion model. The performance of the consistency model does not seem to be better with larger $N$. It would be good if the authors could elaborate on further improving the performance of the consistency model in the offline setting.
> >
> > > hyperparameter tuning
> >
> > Cannot agree with that. As the authors comment, it is computationally heavy to find the best hyperparameter. My concern is, how can we quickly find the best hyperparameters in some new tasks? If the consistency model can exhibit robustness to some hyperparameters, this concern can be mitigated to some extent. While this seems to be lacking in the current version.
> >
> > > different baselines for different domains
> >
> > The authors can find recent baselines on Adroit in recent papers, e.g., [2]. It would be better if the authors could run baselines on kitchen datasets. I also do not want to blame the authors too much on this point and I can totally understand that training consistency model itself can be computationally heavy.
> >
> > > gradient norm
> >
> > I think the influence of gradient norm ought to be included in the paper. It is somewhat strange that you tune this on kitchen datasets but not on other tasks.
> >
> > All in all, I am still somewhat negative about this submission. However, I decided to stand neutral during voting, and would not be frustrated if this paper got accepted.

---

> ### Author Response · Authors · 2023-11-22
>
> Thanks a lot for your constructive feedback and insightful questions.
>
> > why on many offline and offline2online tasks, the consistency model underperform the diffusion model, while on some online hard tasks, the consistency model seem to be better?
>
> We are very willing to share insights about this phenomenon in future work with more extensive experiments. Our current conjecture is the deficiency within the offline pre-trained models and a straightforward usage of offline models with online data. Future plans involve better ways combining pre-trained models and online data, but it can be beyond the scope of current paper since it mostly focuses on the effectiveness of consistency model from offline data.
>
> > statistical significance
>
> We totally agree with the reviewer that the statistical significance is essential for RL. [1][2] are methods not included in current paper but can definitely be added as baselines. However there are lots of methods in this domain and we can only list some most representative ones with limited pages.
>
> > gradient norm
>
> The gradient norm is tuned for tasks that are observed to show less stable training progress.
>
> We generally appreciate your comments. However, some additionally required experiments like evaluating the robustness of the models against each hyperparameter, or running some baselines on certain tasks for providing the statistical significance can take lots of additional efforts and computation. If our response has addressed some of your concerns, we would highly appreciate it if you could re-evaluate our work and consider raising the score.

---

### Official Review · Reviewer_gzrQ · 2023-10-31

**Soundness:** 3 good
**Presentation:** 3 good
**Contribution:** 3 good
**Rating:** 6
**Confidence:** 3

**Summary:**

This work proposes utilizing the consistency model to learn policies in modeling multi-modal data from image generation, which perform more efficiently than diffusion models. The authors evaluate their models on three typical RL settings: offline, offline-to-online, and online, and experiments show that the consistency policy can reach comparable performances than the diffusion policies while reducing half computation costs.

**Strengths:**

- The paper is well-written, and the experiments are sufficient, which include three different RL settings and four task suites with BC and RL baselines.
- The motivation behind this paper is natural and valuable.

**Weaknesses:**

The author claims the consistency policy is much more efficient than diffusion policies while keeping comparable results. However,
- the results in Tab. 1 and 2 can not support the conclusion in a way, which shows there are some significant drops in some tasks (such as halfcheetah-me, kitchen-xxx) between the Diffusion-BC and Consistency-BC or Diffusion-QL and Consistency-AC.
- Moreover, in Tab. 3, when N = 5, the performance between Diffusion-QL and Consistency-AC is comparable while the time cost is also similar. This indicates that when the denoising step is small, the absolute scores of both methods are good enough. In this case, the consistency model has few advantages, which makes the improvement much more limited.

**Questions:**

(1) The authors claim that "By behavior cloning alone (without any RL component), using an expressive policy representation with multi-modality like the consistency or diffusion model achieves performances comparable to many existing popular offline RL methods.", I'm wondering where is the multi-modality. Is the consistency policy trained for all tasks across all suites? Or does the offline dataset have different successful behavior policies? The author should make it more clear.

(2) In the specific tasks in RL, the scenarios are not abundant and the trajectories are quite similar. If more expressive policy representation can result in better performances, what if using some large pre-trained representation models, can this problem be solved? (E.g. R3M)

---

> ### Author Response · Authors · 2023-11-16
> **Response to review**
>
> We appreciate the insightful comments and questions from the reviewer.
>
> Less comparable performance for consistency models on *halfcheetah-me*, *kitchen-xxx* in Tab. 1 and 2. We admit that there are some tasks showing consistency policies are less performant than diffusion policies, and the computational efficiency is not obtained for free compared with diffusion models at present, as it is an acceleration method without full capability of replacing the diffusion models with a much smaller $N$ yet. However, there are still quite a few tasks showing at least equivalent performances of consistency policies over diffusion policies, like *hopper-medium*, *walker2d-medium*, *hopper-medium-replay*, *walker2d-medium-replay*, *Antmaze* tasks on BC setting, and *halfcheetah-medium*, *halfcheetah-medium-replay*, *walker2d-medium-expert*, *antmaze-umaze-diverse* for actor-critic RL setting.
>
> For the result in Tab. 3, our major claim is that Consistency-AC can achieve a similar performance as Diffusion-QL with a much smaller $N$. Specifically, if we look at $N=2$ for Consistency-AC while $N=5$ for Diffusion-QL, the scores are similar but Consistency-AC saves half inference time (1.84/3.76) and training time (31.94/57.06). This holds for most of the tasks in our experiments, and by default, Consistency-AC and Consistency-BC use $N=2$ while Diffusion-BC and Diffusion-QL use $N=5$ for results reported in paper.
>
> For the first question regarding the multi-modality of the data, the model is not trained across tasks but within each task. Even so, the offline dataset for each task is usually multi-modal since it can be collected by a mixing of behavior policies. For example, the 'medium-replay' dataset in Gym tasks is the replay buffer of a RL policy to reach a medium level performance, which contains samples collected by an evolving policy. In our paper Fig. 5 in Appendix A shows the diversity of the data distribution, although it is not the action distribution given a certain state, the shapes also indicate the complexity of the sample distributions. More details about these tasks are provided in the original D4RL paper. We thank the reviewer for pointing out this problem and will add explanations in the modified version.
>
> For the second question regarding whether large pre-trained representation model will be an alternative of expressive policy,  we believe pre-trained visual representation can definitely lead to improved performance for certain domains of RL, e.g., robotic manipulation, but the improvement can source from different aspects compared with the expressiveness of the policies. Pre-trained visual representation is more helpful for providing image information abstraction with semantic understanding. R3M is encoding images as state representation, while consistency/diffusion model represents the conditional distribution of actions given states, so the representation spaces are not the same. If R3M is modified to encode both images and robot actions, while maintaining a multi-modal model, it might be close to the expressive policies. But for current R3M in imitation learning, if the behavior dataset is multi-modal, and the downstream policy of R3M is unimodal (like Gaussian), we may not expect it to capture the multi-modality.

---

### Official Review · Reviewer_259J · 2023-11-01

**Soundness:** 3 good
**Presentation:** 3 good
**Contribution:** 3 good
**Rating:** 6
**Confidence:** 3

**Summary:**

This paper proposes to use the recent "consistency models" as a way of parameterizing the policy for deep RL.  This is explored in offline RL, offline-to-online RL, and online RL with an actor-critic setup.  The performance is competitive with diffusion-BC despite requiring far fewer sampling steps.  This seems like a strong empirical advance to me, because the speed of sampling is essential for online reinforcement learning.  While the approach here is unsurprising, combining consistency models with RL, this still seems like an important contribution.

notes:
  -Diffusion inference is slow, so there could be value in using consistency models to define the policy, particularly for an actor-critic style algorithm.
  -This paper considers both online, offline-to-online, and offline setups.
  -The policy class is important for RL, especially that it be multi-modal.
  -The paper lays out policy regularization to prevent out-of-distribution actions.

**Strengths:**

-I think that this project is very critical for the success of RL, as insufficiently rich policy classes are a major limitation.  Additionally, in RL it is important to draw as many samples as possible (either in a model or in the environment) so evaluating the policy quickly is critical.  So I think this work will have a lot of impact.
  -I think it's also particularly impressive that the paper shows success in online RL, because in online RL it is important that the policy perform well even when it's imperfect (i.e. early in the training process).  Whereas in offline-RL, we could imagine that the policy only needs to perform well near the end of the training process.  We indeed see that the consistency model outperforms the diffusion model in the purely online setting (Figure 4).

**Weaknesses:**

-The idea of using consistency models as RL policies is fairly intuitive and not terribly surprising.
  -On the harder tasks like Kitchen and Adroit, there is a significant gap with Diffusion-BC baseline.

**Questions:**

-Have you thought about also using the consistency model as the "model" in the RL sense, i.e. to learn p(s' | s,a)?  If so, do you see any interesting challenges or opportunities there?

---

> ### Author Response · Authors · 2023-11-16
> **Response to review**
>
> We appreciate the insightful comments and questions from the reviewer.
>
> For the performance gap on Kitchen and Adroit tasks, we show that the computational efficiency is not obtained for free compared with diffusion models. The dataset of the two domains has poor coverage (Adroit) or long-term trajectories (Kitchen), which makes them harder to learn. How to improve the performance of consistency models on these tasks requires further investigation.
>
> We appreciate the idea of using a consistency model as a transition model instead of policies and it is definitely worth trying. It could be more interesting to see the consistency model being used for sequence prediction like Decision Diffuser, which requires more expressiveness and multi-modality. In that case the improvement of computational efficiency can be even more significant. We generally expect the significance of leveraging the consistency model for RL to be larger in more complicated models or tasks, and current experiments are showcased on standard D4RL baseline.

---

### Author Response · Authors · 2023-11-16
**General response to all reviewers**

We first would like to thank all reviewers for their efforts and instructive feedback. Some of the questions are explained and reflected in the modified version of the paper, which are highlighted in red in the modified paper.

We appreciate that the values of the paper are recognized by reviewer **259J** that: 1. Consistency model is a strong empirical advance since sampling speed is essential for online RL. 2. This work will have a lot of impact. Reviewer **gzrQ** mentions the experiments are sufficient. Also, reviewer **JCUm** recognizes that the paper is the first work that introduces the consistency model into the offline RL setting, online RL setting and offline2online setting.

We will emphasize several common concerns here:

* Regarding the novelty and an intuitive idea of consistency models as RL policies (by reviewers **259J** and **JCUm**). We admit that applying the consistency model as policy representation for RL is intuitive. The convenience of the deployment of the consistency model verifies its general usage as a multi-modal model. On one hand this is a good thing since people usually prefer general yet simple-to-use models, on the other hand this may not meet the novelty expectation of research since not too many additional tricks need to be added. However, we still think it is valuable to propose the usage of more efficient yet expressive models for RL, to satisfy the needs of larger-scale experiments and heavy online interactions in the end, even if the current method is slightly less performant compared with the diffusion models (which is already shown in original consistency model paper). We also want to mention that during the submission of the paper, there are already improved techniques [1,2] for consistency models, which are very promising for further improving the current results with minor changes. This paper will be inspiring for researchers to follow this road.

* We also notice that Tab. 3 may be interpreted in a different way by reviewers (**gzrQ**, **JCUm**) from what we expect and we have modified the presentation of this table to make it clearer. We want to highlight that in experiments for all environments and settings (not just *hopper-medium-expert-v2* here), by default, Consistency-AC and Consistency-BC use $N=2$ while Diffusion-BC and Diffusion-QL use $N=5$ instead of using the same $N$. By reading the result in Tab. 3, it indicates a reduction of around half of inference time instead of a minor reduction, since the consistency model saturates its performance with $N=2$ without the need of a larger $N$. By default in our experiments, consistency models use $N=2$ and diffusion models use $N=5$ as specified in Sec. 5 first paragraph.

* We want to emphasize that the proposed consistency models for RL policies are not expected to fully outperform diffusion models in scores, which also holds for the computer vision domain. Therefore we did not claim that consistency policies are always comparable or better than diffusion policies on scores in each Empirical findings 1, 2 and 3. We fairly reported our discoveries by using consistency models in replacement of diffusion models for different RL settings. What we want to address is the trade-off between computational efficiency and model accuracy, and we never claim that the consistency-based policies are always better or comparable than the diffusion-based policies in terms of performances in offline RL. However, we claim that consistency-based policies are better in terms of computational efficiency, and this can be very critical for RL with environment interactions or in complicated tasks, as also pointed out  by reviewer **259J**.

The other comments and questions by each reviewer will be replied in detail correspondingly, we will try to explain and remove concerns for the reviewers. If reviewers still have any question, we are glad to answer during the rebuttal period.

References:

[1] Song, Yang, and Prafulla Dhariwal. "Improved Techniques for Training Consistency Models." arXiv preprint arXiv:2310.14189 (2023).

[2] Kim, Dongjun, et al. "Consistency Trajectory Models: Learning Probability Flow ODE Trajectory of Diffusion." arXiv preprint arXiv:2310.02279 (2023).

---

### Comment · Area_Chair_AEDf · 2023-11-21
**A borderline case requiring discussion**

Dear Reviewers,

The authors have addressed each of your comments and have made corresponding revisions to their paper. I kindly request that you review these changes and provide feedback on whether your concerns have been adequately addressed. Please also consider adjusting your evaluation scores if you find it appropriate.

Thanks,

AC

---

### Meta-Review · Area_Chair_AEDf · 2023-12-08

**Metareview:**

The primary aim of this paper is to introduce consistency models as a versatile and efficient policy class for reinforcement learning (RL). The proposed Consistency-Actor Critic (AC) algorithm closely resembles Diffusion-QL, with the main distinction being the replacement of the diffusion model-based policy in Diffusion-QL with a consistency model-based policy. Diffusion-QL, designed to introduce a flexible policy capable of capturing multimodalities and dependencies between different action dimensions in offline data, has demonstrated effective performance in offline RL. The paper demonstrates that Consistency-AC can achieve competitive performance using only two sampling steps, in contrast to Diffusion-QL's default of five sampling steps. This suggests that Consistency-AC can enhance sampling speed, albeit with a trade-off of slightly degraded performance.

**Justification For Why Not Higher Score:**

The level of novelty is constrained as it builds upon the previously published Diffusion-QL paper by substituting the Diffusion model with the Consistency model and constructing a paper around this modification.

**Justification For Why Not Lower Score:**

Firstly, the practical significance of reducing the number of sampling steps from five to two is evident.

Secondly, the experiments outlined in the paper emphasize a clear advantage in pretraining a flexible policy, whether using Diffusion-QL or Consistency-AC, followed by continued online training, as opposed to relying solely on online training. Notably, the strategy of offline pretraining followed by online finetuning has not been previously explored in the context of the Diffusion-QL paper.

---

### Decision · Program_Chairs · 2024-01-16

Accept (poster)